# Evaluation of a program targeting sports coaches as deliverers of health-promoting messages to at-risk youth: Assessing feasibility using a realist-informed approach

Emelien Lauwerier[1,2]*, Esther Van Poel[1], Karen Van der Veken[1], Kaatje Van Roy[1], Sara Willems[1]

1 Department of Public Health and Primary Care, Ghent University, Ghent, Belgium, 2 Department of Experimental-Clinical and Health Psychology, Ghent University, Ghent, Belgium

* Emelien.Lauwerier@ugent.be

**Data Availability Statement:** All relevant data are within the paper.

## Abstract

Unequal access to health promotion resources and early prevention services is a major determinant of health inequity among youth. Initiatives that improve the access to and adoption of health promotion messages are important undertakings, e.g., sport. Sport-for-development (SFD) programs are seen as valuable delivery tools, in which coaches are used as change agents to increase health awareness and behavior among at-risk youth. The delivery of such messages requires specific knowledge and skills that can be attained through training; however, the effectiveness of such training requires assessment. In this study, we evaluated the feasibility of such a training program for SFD coaches using process evaluation from a realist perspective, and views from multiple stakeholders, among other sources. We also clarified the inner workings of the training and investigated how context shaped the training outcomes. Increased health awareness and a sense of responsibility from acting as a role model for at-risk youth were among the perceived training outcomes. Building a safe environment for learning, engagement, and bonds of trust increased the confidence to learn, and resulted in a sense of critical self-reflection and self-development of SFD coaches towards health and prevention messages. Importantly, the unique situations (or context) of SFD coaches and SFD in general presented challenging variables, e.g., a precarious life history or living conditions, mental health issues, or low educational skills, that hampered the impact of the mechanisms put in place by the training. Here, we present a process in which the development of the 'right mind-set,' engagement and bonds of trust, in combination with the right settings are key elements for SFD coaches to learn how to convey health-promoting messages and take responsibility as role models for at-risk youth.

## Introduction

Childhood and adolescence are important life phases for the development of healthy adults. However, health is not equally and fairly distributed. In socioeconomically disadvantaged

**Funding:** The study was funded by Flanders Innovation & Entrepreneurship under Grant number IWT 150060. The content is solely the responsibility of the authors and does not necessarily represent the official views of the funding agency or other project partners. The funders had no role in study design, data collection and analysis, decision to publish, or preparation of the manuscript.

**Competing interests:** The authors have declared that no competing interests exist.

groups, inequalities in health already emerge in early life, e.g., with higher risks of low birth weight, prematurity, and infant mortality [1–3], and persist throughout childhood and into adolescence. Children and young people (CYP) in socioeconomically disadvantaged circumstances suffer higher rates of poor mental well-being, and longstanding illnesses, such as obesity and asthma [1, 4]. A large-scale study involving randomly sampled schools in 37 countries across Europe and North America revealed that adolescents with a lower family wealth index showed a higher prevalence of daily health complaints, such as pain or mental health issues, compared with adolescents from wealthier families [5]. The conditions during early life not only affect childhood health but can also have long-term effects on adult health. For example, obese children are more likely to develop obesity as adults [6], a condition that is associated with an increased risk of a number of serious health conditions, such as cardiovascular diseases, type 2 diabetes, and cancer [7]. Because of this, inequalities in the socioeconomic circumstances during childhood contribute to the health inequalities seen in adulthood.

Inequity in access to health services, such as early health promotion and prevention, is a major determinant of health inequity among youth [8]. Prevention is key to improving health outcomes by targeting unhealthy behavior such as smoking, poor diet, physical inactivity, or alcohol use. All CYP should have access to prevention services because unhealthy behavior has an early onset [9–13], tends to persist into adulthood [14], and can, therefore, increase the risk of morbidity and mortality at a later age [15]. Inequities in long-term health can be reduced not only by improved access to health promotion and prevention services but also by the tailoring of these services to the circumstances of CYP, among other initiatives [8].

Sport-for-development (SFD) may provide a setting through which health promotion and prevention messages can more easily reach and be adopted by hard-to-reach populations. SFD initiatives use sports as a vehicle to tackle other issues, such as education, employment, community involvement, health promotion, and prevention [16]. SFD is a relatively new sector and the inner workings of SFD initiatives have been actively studied since the late 1990s [16–18]. The successes of health promotion initiatives are linked to the barriers that poor and marginalized communities face, such as limited access to health services, a lack of physical access, no affordability, and low acceptability due to specific cultural norms. However, because such activities are typically low-threshold, financially accessible, and locally organized in specific– and often urban–neighborhoods, some of these important barriers are lifted. In addition, SFD coaches play a vital role in the successes of delivering the activities. Because coaches have regular contact with CYP, they have unique opportunities to build trusted relationships that will enable them to facilitate positive changes in the behavior and attitudes of CYP [19, 20]. Coaches are also perceived to be proximate and positive role models for health behavior change [21]. There are examples of health prevention programs delivered by sport coaches [22–24], albeit not in the context of SFD and not specifically targeting action in vulnerable groups. In addition, the effectiveness of the interventions depend on the extent that coaches manage to deliver appropriate knowledge, install appropriate attitudes, and apply appropriate skills regarding the health promotion activities to be delivered [25]; which is especially challenging within SFD. SFD coaches also introduce a range of different skills and personality profiles into SFD initiatives, given the different roles and skills required to coach at-risk youth. Some of these roles include being a trustful friend, a liaison officer between the youth and youth organizations, and a technical coach to develop the sportive capabilities of some individuals [26, 27]. In Flanders, Belgium, organizations often combine these different roles by recruiting different profiles. For example, such scattered workforces can consist of mid- to highly educated coaches that possess sportive or pedagogical degrees (or both) as well as experienced experts that often do not have degrees and have low educational skills.

To enhance the evidence-base regarding the improvement of health outcomes in vulnerable populations by SFD coaches, it is important to gain a detailed understanding on what makes coaching successful in this specific context. To achieve this, we conducted a process evaluation built on realist theory [28]. Realist theory recognizes that many variables operate at different levels, and which account for differences in program effects. It is imperative to acknowledge that interventions do not necessarily work for everyone, because of differences among people and the contexts that they are embedded in. It is, therefore, vital to clarify which elements influence the effectiveness of programs (i.e., mechanisms of impact) and which external variables (i.e., context elements) may hamper or facilitate their impacts [29]. This knowledge is imperative to develop optimal complex interventions and, thus, contributes to decision-making regarding the implementation of interventions on a larger scale [30].

Here, we conducted a process evaluation of an SFD training that targeted coaches to improve their knowledge and skills for the transfer of health promotion messages to at-risk youth. We explored the feasibility of the training program (also hereafter referred to as the 'intervention'), and were also interested in the 'theory-of-change' (i.e., how and when it works) underlying the training. The training was previously developed (for a more detailed overview of the program, see [31]), and was implemented and evaluated in a specific case setting, i.e., the community sport activities of a middle-to-large city in Flanders, Belgium. The aim of this study, was to gather data from multiple stakeholders within this specific context, including SFD coaches, staff, and local policy makers, to enable an in-depth analysis of the feasibility of the training. The study objectives were to determine how to effectively train coaches to ensure their viability as deliverers of health prevention messages to at-risk youth populations, and clarify which elements external to the training and specific to SFD programs can lead to the successes or failures of such training. This study obtained valuable insights that inform researchers and policy makers on the training required to ensure coaches are viable deliverers of health prevention messages to at-risk populations.

## Materials and methods

### Participants and recruitment

In Flanders, Belgium, ~22% of Flemish municipalities provide SFD activities [32]. These activities are usually subsidized by local governments, and are mostly directed towards the social inclusion of disadvantaged groups, in particular vulnerable CYP [33]. In Bruges, the setting of the present study, SFD initiatives operate under the supervision of Bruges' Public Centre for Social Welfare, which coordinates the social services in the city. Its activities run within the four most deprived neighborhoods in the city, which are characterized by high numbers of single-parent families, children with learning difficulties, unstable accommodation, and low employability.

In February 2018, all SFD coaches (n = 8) and the pedagogical and sportive staff (n = 3) that deliver SFD programs in Bruges were invited to attend a training program. All invited SFD coaches were male, with a mean age of 30.4 years (age range = 25–43 years). The study received ethical approval from the Medical Ethical Committee of University Hospital Ghent (B670201835740). All SFD coaches and staff provided written informed consent (that complies with the details stipulated in the PLOS consent form) to participate in the research study.

### Description of the training

The current study forms part of a four-year (2016–2019) research project, named CATCH–Community sports for AT-risk youth: innovative strategies for promoting personal development, health, and social CoHesion–which is aimed at exploring 'why, how, for whom, and

under which circumstances' community sport activities enhance the personal development, health, and social cohesion of at-risk youth. The outputs of previous studies resulted in the development of a program theory and its further refinement by clarifying why, how, and when community sport can promote health, in terms of the mechanisms and context factors of improved health outcomes of at-risk youth. These results are described elsewhere [34, 35], and a training program was developed for community coaches based on the insights of the program theory [31]. A brief overview of this training program is needed to enable the interpretation of the current study, and it is, therefore, described below.

The training program aimed to: (a) increase the awareness and knowledge of coaches on the effects of health behavior on overall health, well-being, and sport performance (e.g., smoking, physical inactivity, poor dietary habits); (b) increase their awareness and knowledge on the mechanisms to promote the health of CYP; and (c) introduce tools and skills to encourage CYP to participate in community sport activities and adopt a healthier lifestyle. To do so, the training program covered topics such as health promotion, healthy living, positive coaching, communication, team dynamics, and conflict. Several strategies were also adopted, including group sessions moderated by one or two tutors with game-based activities, theory and information provision, reflection and discussion exercises, and peer observations. In addition, several individual sessions were planned between and after group sessions. After their first series of four group sessions, each SFD coach had two individual sessions with a job coach, with whom they were already acquainted and had regular encounters regarding their personal (work) trajectories. The aims of these sessions were to encourage elaborated thinking regarding their health status and personal health goals, and the setting of personal action plans regarding their health. At the end of the program, the job coach planned to have at least one follow-up session to discuss the progress of the coaches towards their own healthy living, as well as their concerns or problems in applying skills to promote health among the youth attending the community sports activities. The training aimed to promote change through the application of experience-based and active learning methods, such as raising awareness, guided practice, and skills development, among others. The step-by-step development of the training program, including the links between the methods and their application, is described in detail elsewhere [31].

The training was delivered over several months (between March and December 2018). The training program was co-created and implemented via the close collaboration of the researchers, and the staff and key stakeholders from the intervention site in Bruges. At least two tutors, either the researchers (EL, KVDV) or the staff (LG, RS, NVB), or both, moderated the group sessions. The present study focused on evaluating the feasibility and implementation of this training.

### Evaluation design and measures

Multiple process measures were integrated into the design based on the Medical Research Council (MRC) guidelines for evaluating complex interventions [29]. We measured the intervention feasibility (i.e., reach, dose, fidelity, acceptability), but also assessed the supposed theory-of-change, i.e., which factors in the intervention regarding 'how' and 'under which circumstances' led to which effects (e.g., what changes occurred over the course of the intervention that led to some of the observed impacts, what elements appeared to make a difference, etc.). In particular, we used a realist-informed process evaluation (and not a realist evaluation per se) to explore this 'theory-of-change.' Realist evaluations consist of a set of methods and are characterized by an iterative set of stages that involve developing, testing, and refining a theory. Our analysis approach was more inductive, open-ended, and sought to document how

the training resources promoted the changes among participants that led to the observed outcomes, under specific circumstances [36]. We initially assessed the empirical data, and then tried to identify hidden processes and develop ideas using a realist data analysis process called 'retroduction' [37] to clarify what aspects of the training worked for SFD coaches, why the training was effective, and under which circumstances the outcomes occurred. Our identification of hidden processes was informed by a pre-developed program theory on the mechanisms and influencing context factors of improved health among at-risk youth [34]. The current study is best conceived as an in-depth refinement of one of the key-influencing factors within this program theory, being the application of motivational coaching and the installation of positive group dynamics by coaches to increase the health outcomes of at-risk youth. Central mechanisms of impact are experiential learning, incremental responsibility-taking, and reflexivity that may, under the right circumstances (e.g., participants feeling safe around others), beneficially influence the health outcomes. These processes and some of the contextual variables may be transferable to the coach training situation regarding health promotion within this same setting of community sports because the local coaches appear to have similar at-risk backgrounds compared to the youth themselves. Therefore, the mechanisms and contextual variables within the program theory provided a good starting point from which interpretations and hidden influential factors could be inferred from the data in the present study. A visual scheme depicting a 'translation' of mechanisms of impact, contextual variables, and outcomes is presented in Fig 1.

We adopted a summative approach to data collection from multiple stakeholders (SFD coaches, staff, local policy makers) and used different measures to obtain the range and depth of data required (see Table 1).

Process evaluation measures included document logs, direct observations of intervention delivery, session evaluation questionnaires (for SFD coaches and staff), semi-structured interviews with SFD coaches and staff, and a focus group with staff and key stakeholders involved in local or national sport and recreational (community) activities, and local policy.

**Document logs.** Information regarding the intervention delivery was recorded in a logbook, including the number of participants in each session and in the individual sessions. Communications (including emails, phone calls, and face-to-face discussions) among researchers and staff delivering the intervention modules and sessions were also logged. These document logs were used to assess the intervention reach and dose.

**Direct observations of intervention delivery.** To explore the reach, dose, fidelity, and acceptability of the intervention, participatory observations of group sessions were conducted (n = 10 in total). At least one researcher was present at each group session to take observational notes. The observer had a general idea of what may be salient but aimed to keep an open mind. The observations were, therefore, unstructured and unfocused, and the narratives were written down with the aim of documenting as much information as possible. These narratives included detailed information on how the activities were delivered to and received by SFD coaches and staff.

**Semi-structured interviews.** Individual face-to-face interviews with participating SFD coaches and staff (n = 8) were used to explore the theory-of-change and acceptability of the intervention. Interviews took place within a maximum of 8 weeks after the end of the intervention. All interviews were held in person by the third author in a private room at the SFD setting. This researcher had content-related experience which gave breath to the data collection. Due to this experience, a self-reflective stance was adopted during interviewing, and, having extensive methodological expertise, a general openness and curiosity about the interviewees' experiences was established. Interview topics covered all components of the intervention, including the perceptions of the intervention modules and sessions, their supposed impacts,

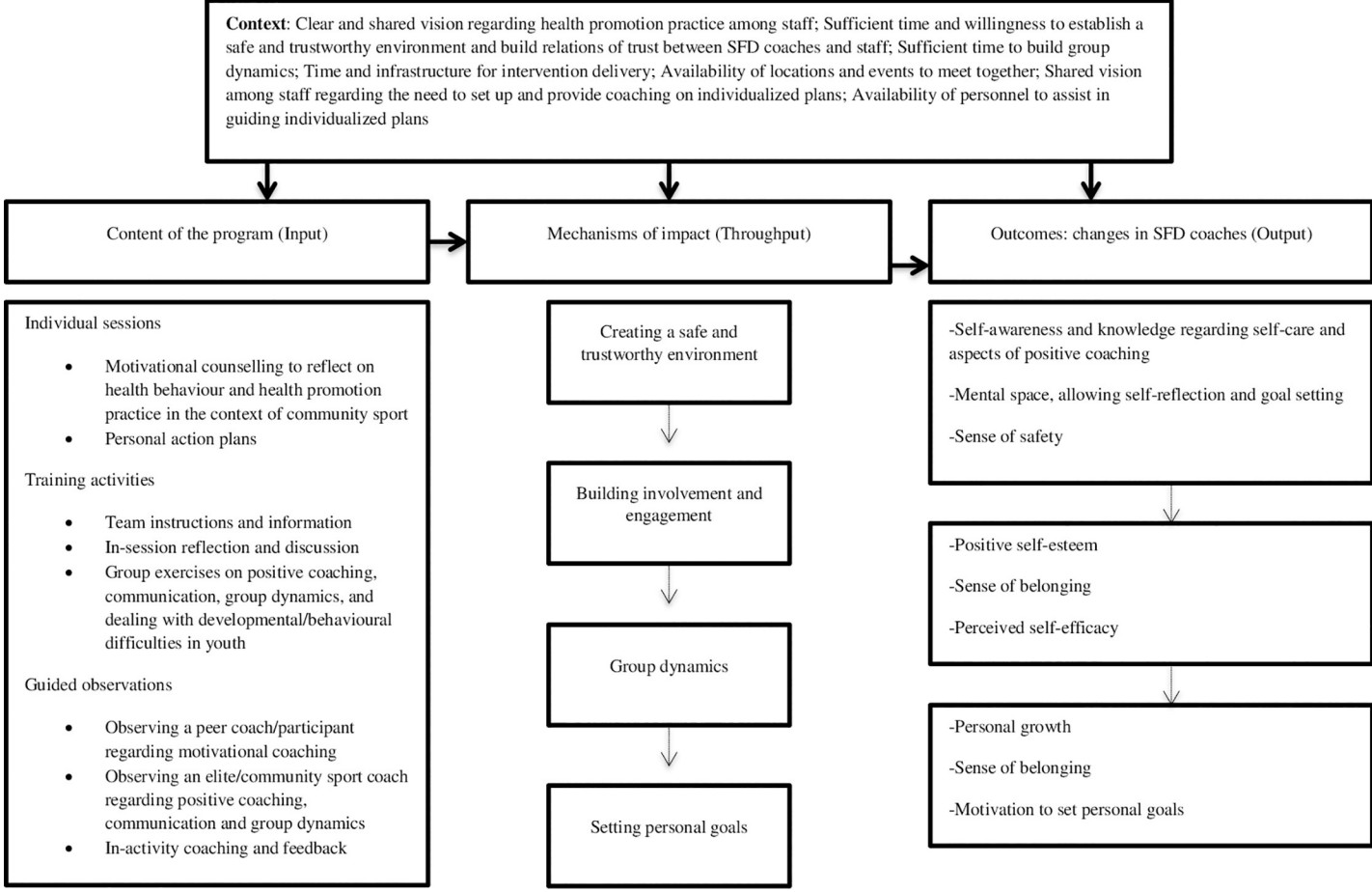

**Fig 1. Theory-of-change underlying the training targeting sport-for-development (SFD) coaches to deliver health promotion messages.**

**Table 1. Data sources used to assess the feasibility of the training, including sampling details, time frame of data collection, and aspects of feasibility assessed for each data source.**

| Data source | Sample | Date of data collection | Feasibility aspect assessed | | | | |
|---|---|---|---|---|---|---|---|
| | | | Reach | Dose | Fidelity | Acceptability | Theory of Change |
| Document logs | (Intermediate) reporting between staff (n = 3) and researchers (n = 2) | Mar 2018 –Dec 2018 | X | X | | | |
| Observations | 10 sessions; SFD coaches and staff (n = 5–8) | Mar 2018 –Dec 2018 | X | X | X | X | |
| Interviews | SFD coaches and staff (n = 8) | Feb 2019 | | | | X | X |
| Focus group | SFD staff, researchers, and stakeholders involved in local or national sport and recreational (community) activities, and local policy (n = 8) | Dec 2018 | | | | | X |

SFD, sport-for-development; Reach, the intended audience came into contact with the intervention and how; Dose, the quantity of interventions implemented; Fidelity, the intervention was delivered as intended; Acceptability, the intervention was acceptable to the users; Theory of change, how the delivered intervention produced change and how the context affected the implementation and outcomes.

working ingredients, and the facilitating and hindering context elements. In addition, the staff were asked about the delivery of the sessions and their perception regarding the engagement of SFD coaches during the intervention. All were given the opportunity to comment on topics they believed were not covered in the intervention. All interviews lasted 30–60 min and were audio recorded.

**Focus group.** A focus group comprised of staff that participated in the intervention as well as key stakeholders (i.e., local policy makers, representatives of sport and recreational organizations within the region of Bruges, and a representative of an expert organization on community sports in Flanders) (n = 8) was used to explore the theory-of-change of the intervention. The focus group lasted 60–90 min, and was audio recorded. Two researchers (EL and KVDV), both of whom had different content-related and methodological expertise, collected the data. The perception of the focus group members regarding the intervention, their supposed impact and working ingredients, facilitating or hindering context elements, and suggestions for broader dissemination were explored.

### Data analysis

The transcripts of the focus group and interview recordings, narrative reports of session observations, and document logs were imported into NVivo version 12 and subjected to thematic analysis [38]. This process involved familiarization with the data (reading and re-reading) and assigning broad thematic codes (EVP). Some of these codes were pre-defined from topics covered by the logic model of the intervention (see Table 1) and the MRC framework for evaluating complex interventions [29]. Researcher triangulation was applied. A second qualitative researcher (KVR) read all the transcripts and performed the coding separately. Coding was discussed and confirmed by the research team (EL, EVP, KVR). Subsequently, the team derived broad, higher order themes from specific codes, and a descriptive summary was written based on recursive engagement with the data. Although the themes derived were mainly data-driven, the logic model and MRC framework were guiding frameworks used to structure the data. The researchers continuously reflected on the way these deductive themes should be integrated with the data. For confirmation, two researchers that had not yet been involved in the analysis, read the synthesized text and interpretations and checked these against the data and quotes (KVDV, SW).

## Results

### Feasibility

**Reach.** The intervention program was delivered and assessed within the context of a specific SFD organization. The intervention aims were co-created by researchers and staff, and developed within the community sports practice activities and individual trajectories of SFD coaches. The intention was, therefore, for the intervention to reach all SFD coaches and staff. Despite this intention, only two of the SFD coaches attended all sessions. Reasons for non-completion included: illness, conflicting duties, and personal or psychosocial issues.

**Dose.** All 10 group sessions were delivered over the course of the intervention period. Group sessions were conducted at 2–3-week intervals, and with a 2-month summer break, which enabled observational activities and individual sessions to be conducted between group sessions. The intervals between sessions, however, tended to vary and become lengthier (maximum 1-month interval) over the course of the intervention period. Despite the initial plan, individual sessions often did not take place between group sessions and after the intervention program. Between group sessions, only four of the eight SFD coaches attended one individual session, instead of the planned two sessions. In addition, none of the coaches attended

individual follow-up sessions. The discrepancies in the planned and attended sessions may be a result of insufficient time during the intervention period or the need to lower the intensity of the intervention as a whole; thus, resulting in the choice to focus primarily on the group sessions.

**Fidelity.** Overall, the session content was delivered as planned, but we also observed disparities. Activities and exercises were omitted to a minor extent. In most cases, however, task assignments that were scheduled between sessions were not fulfilled. Reasons were explored further during reflection sessions with staff that were organized intermittently across sessions, as well as during interviews with SFD coaches. Changes were made because of reasons related to the intervention setting, such as the flow (e.g., too many absent participants) or organization (e.g., delays in session timing, distractions leading to the early concluding of sessions, logistical issues). Other modifications to the sessions were made by the SFD coaches because of an 'information overload,' because the sessions were too demanding when combined with regular activities, and because of an unfamiliarity with session activities, such as open reflection moments. The take-home assignments were abandoned because the observation coaches needed more time and guided instruction to fulfill these. The SFD coaches did not opt to spend more time on the assignments outside of the group sessions for the same reasons listed previously regarding the low attendance of the individual sessions.

**Acceptability.** Participants perceived the intervention as important, satisfactory, and educational. The intervention was not perceived as something additional, but rather as something that supported their current work.

> *"(. . .) I also think that it* [the intervention] *did them* [the SFD coaches] *good as they ultimately looked forward to the intervention sessions (. . .). Yes, I felt this was the case and they* [the SFD coaches] *also felt it was important (. . .)* [name of SFD coach] *said: 'I won't take time off then because the sessions are happening then.'"* (Staff member 1)

> *"(. . .) because of it* [the intervention] *being so strongly linked to our own needs and what we also want, it is very nice. Genuinely supportive. We saw it as something supportive instead of extra work."* (Staff member 1)

**Overall.** The above findings offer a general overview regarding a diversity of indicators related to feasibility. However, it is important to gain an in-depth understanding of why the training was perceived as acceptable, but was not able to reach all coaches, and the reasons for the apparent disparities regarding dose and fidelity. The following section aims to offer further insights to clarify which training processes led to its success, and identify influential contextual variables by exploring the underlying theory-of-change as perceived by coaches, staff, and other stakeholders.

## Towards a supposed theory-of-change: Description of perceived mechanisms of impact and influential contextual variables

**Creating a safe and trustworthy environment.** Our findings highlighted that one of the key training processes involved the extent to which the learning activities were set up within a safe and trustworthy environment. Several strategies were mentioned in this context, such as time adjustments (e.g., organization of breaks and session intervals according to needs and demands) and setting adaptations (e.g., arrangements of tables and chairs). Also, the SFD coaches explicitly praised the openness and willingness of researchers to ask for feedback regarding the session content and methods used. This seemed to beneficially enhance the

sense of self-efficacy of coaches to re-think and adapt health promotion practices and coaching.

> *"So, I do think that was something for them* [the SFD coaches] *to look forward to. Of course, for some, the idea of education and training is difficult, but I do think that, because we approached it in a very accessible way and because they* [the SFD coaches] *gave a lot of input, it did go better after a while."* (Staff member 1)

Besides openness leading to an increased sense of self-efficacy, it also resulted in some difficulties. For example, the non-mandatory nature of take-home assignments between sessions led to few or none of the coaches fulfilling these assignments; and because SFD coaches were not 'obliged' to take part in sessions, only a few took part in the entire series of sessions, with one individual not returning after only a few sessions. The low-threshold approach may also have led to disparities in the session content compared to what was planned beforehand. In cases where already vulnerable coaches had to combine the training with regular job activities, sessions tended to be very easily perceived as work overload and of too high intensity.

> *"Don't get me wrong, I think what you guys are doing is good, but it's too reasoned and structured. We talk about the emotional tank of youths, but we* [the SFD coaches] *also have a tank (. . .)"* (Coach 1)

One staff member called it 'a door swinging both ways'. This member made the parallel with a project targeting at-risk youth and, as such, tried to demonstrate the importance of the unconditional nature of activities, also when targeting coaches.

> *"The project is in collaboration with* [partner name] *and they have received subsidies for about a year or a certain amount of time to–I think–guide young people to help them get work, but for some young people, work isn't really an immediately achievable goal, but they work one step at a time and one of the things that they try and achieve is to motivate sports participation and that's something that is noncommittal, but is an important factor in their trajectory (. . .) With those young people, they want to link the advantages of, for instance, sports participation to certain skills that are also required in their work, for example arriving on time, committing to something (. . .) that's how they want to offer some structure in the daily lives of young people, which in itself is really positive. But on the other hand, we noticed that if it is compulsory, it reminds them too much of school, and that it also works in a rather demotivating manner. Thus, it is a double-edged sword."* (Staff member 1)

**Building involvement and engagement.** Another main process was the extent to which coaches were actively involved and engaged. More specifically, several strategies were described that were linked to involvement and engagement, such as the use of recognizable material, the use of fun activities in combination with theoretical content, the way in which the session content was delivered (e.g., enthusiasm, pace), and the various natures of session activities (e.g., a visit to another community sport practice).

> *"They* [the SFD coaches] *realize the relevance of it because it is really linked to the stuff they do every day in practice, and it is, therefore, very rewarding for them."* (Staff member 1)

Breaks were also crucial between activities; affording participants opportunities to have mental breaks and thus enable them to further engage with the training.

*"I thought it was positive that they* [the researchers and/or staff delivering the training] *provided enough intervals during the sessions, alongside the practical and active exercises. It gave the participants a break and was beneficial for aiding concentration. Too many breaks can sometimes be seen as time-consuming, but to me the moments of relaxation led to a more active participation during the sessions and also made it easier to absorb the material that was presented."* (Staff member 2)

SFD coaches viewed their own past experience in community sports (as children and adolescents) as an asset. Active participation and the introduction of their own experiences contributed to the vividness of health promotion practices, coaching attitudes, and skills to be learned.

[Regarding the content of a particular session] *"(. . .) staff members were allowed to include cases regarding recent or past experiences surrounding the theme of conflict (. . .) they gave very nice examples and brought in theory and you could talk about it with your own experiences. It was also valuable for the members of staff to be able to make the connection, yes, and to express the theory in a very informal manner (. . .)."* (Staff member 1)

Also, active participation and learning through experience increased the awareness of the SFD coaches that they are important role models for at-risk youth, which influenced them to think and act differently regarding health topics. This shift in thinking was clearly observed by staff, and was also determined to be an important mediating outcome and powerful lever to potentially affect change among at-risk youth.

*"Yeah, I think a general idea of also deepening awareness, like 'ok it's not just about finding a fun game and explaining it, but my role is actually bigger. I am important to the young people, to my group, my attitude is important.' Being aware of the smaller things that can make the difference."* (Staff member 1)

[Regarding coaches acting as role models] *"(. . .) You should not underestimate it, I think, for example one of our colleagues isn't the most sporty and he has been made responsible for (. . .)* [name of a project] *but he himself used to be in the same situation and that is actually why he understands it very well and, in a way, it makes him a good role model (. . .)."* (Staff member 2)

At the same time, however, a discrepancy arose between having to act as a role model and feeling able to be one. For example, when the behavior of current coaches was in contradiction with the behavior to be promoted among youth (e.g., smoking vs. preventing youth from smoking). This discrepancy highlights the possibility that personal vulnerabilities can lead to coaches having uncertainties or doubts of whether they will be able to live up to being a role model; thus possibly hampering the link between learning through training and a shift in thinking or the awareness of being a role model for at-risk youth.

*"If you know that you don't always do a good job yourself and then you have to try and convey it to young people or to the participants, I think it's tricky. That might be my character, I do smoke now, eh."* (Coach 1)

*"I think it was also confronting for some people. Like because they also thought 'ok yeah, I'm a role model again and do I live up to being a role model or not,' and that some of them were occupied with doing that."* (Staff member 1)

Several other outcomes that were enumerated and linked to building involvement and engagement included an increased awareness, an improved knowledgebase, and a stronger motivation regarding (new) health promotion practices. One example is the learning of new concepts that gave coaches the language to discuss and explain the things that they were already doing (e.g., coaches already knew the importance of not putting too many demands on youths, but the concept of 'an emotional tank' clarified the concept further, and gave them the language needed to discuss these issues with youths as well as with staff and fellow coaches), and contributed to the motivation of coaches to continue their coaching, for the sake of the health and well-being of youths.

> *"I actually always keep track of the emotional tank. I give a lot of compliments because it helps. When something is fun, then they can handle it better. It helps. If you only say something when they are doing something bad, but never when they are doing something right, then the children's reservoirs are emptied."* (Coach 2)

Also, engagement was perceived to result in increased self-monitoring of current coaching practices. As a result, some SFD coaches specifically mentioned that they had become aware of what they had already done and achieved, leading to increases in overall self-efficacy.

> *"Yes, in general I found that it was a good refresher. Also, with regard to the activities and games we did, we thought 'Aah, we can also do that with the guys.' In fact, every element was a refresher of sorts to me, like 'Ah yes, I hadn't thought of that game for a while, we can also do that and apply it.'"* (Coach 1)

> *"But I do think it* [their confidence] *progressed with an upward trend. Session after session they gained more trust and they could also apply it more in practice, especially after the last 4–5 sessions, which involved positive coaching."* (Staff member 3)

Engagement and involvement also appeared to act as levers for the creation of opportunities to actively practice and rehearse various new skills regarding health promotion practice and coaching. These skill-building moments, deemed to increase the sense of self-efficacy and skills among SFD coaches, were even considered to be necessary elements for the sustained effects of the training.

> *"(. . .) a lot of role playing with a colleague enabled me to know how one can react to certain situations. (. . .) at the beginning, I sometimes didn't know how I needed to react. Now I'm going to do it more easily because I have already done it once."* (Coach 2)

However, staff also mentioned variables external to the training that seemed to hamper its implementation or impact. These were related to the characteristics of coaches as being a vulnerable group themselves, such as suffering from low levels of self-esteem and having (a history of) (mental) health problems. In addition, language problems were apparent in one coach and appeared to hamper their training implementation.

> *"(. . .) some people don't have a lot of confidence and then you notice that if they are confronted with it* [their lack of confidence], *that they do (. . .) you can also say that nobody fits the mold, but that doesn't always work. (. . .) I think that, that may also play a role. Certainly, if at that moment you are also struggling with yourself, then is a theme that is close to you (. . .) it is on your mind."* (Staff member 1)

**Group dynamics and relations.**   In general, several strategies were undertaken during the training with the aim of promoting positive group dynamics, such as the use of games that stimulate team involvement, group communication, and skills. Coaches and staff recognized the benefits of promoting group dynamics for the training to be successful; a process that was easily established within the studied context, in which everyone knew each other adequately well beforehand.

> *"I thought there was already a reasonably good group dynamic here, but those sessions have also helped that. It is also just fun that we take the afternoon off together to follow those sessions, with all of those game elements in between; it made for a good group dynamic."* (Staff member 3)

In addition, the staff highly appreciated the inviting attitude of researchers and tutors regarding the co-creation and co-delivery of the intervention. This process resulted in the growth of constructive bonds and relationships between the researchers and the staff of the particular SFD organization, and seemed to contribute to a positive learning environment. The resulting learning effects, as mentioned by staff, was that such co-working opportunities resulted in deepened reflection regarding their current practices and ways to do better.

> *"(. . .) you think like, yes they* [the SFD coaches] *are not going to like that very much, they're going to find it too school-like. You kind of have to find the courage to handle things in a different way, which is apparently liked and appreciated, and we can also learn a lot from it."* (Staff member 1)

**Personal health monitoring.**   Overall, participants acknowledged the benefits of getting health information and advice on goal setting with the aim of improving their behavior. This appeared to raise awareness regarding the significance of health improvement, as well as the (temporary) increase in health-monitoring activities. The participants also realized that they had to engage more in planning health behavior actions. Both the SFD coaches and staff agreed that the reasons for this was the limited time (within the time frame of the training) for behavioral changes to take place, and no (or safe) opportunities to discuss personal goals within groups.

> *"I think that there should be more guidance for that* [i.e., changing of their own health behavior]. *The seed has been planted but I guess that more is needed to really get started. But the theme was tabled."* (Staff member 2)

> *"I have also tried to cut down* [i.e., smoking] *but after a week the session was in the back of my mind and then I started again."* (Coach 2)

**Fostering discussion and reflection.**   The data revealed the *fostering of discussion and reflection on current functioning* to be one of the most important processes related to the training outcomes, as mentioned by the coaches and staff during the observations and interviews. Staff and coaches also mentioned the importance of reflection being well guided (e.g., through imagery, case-examples, via step-wise instructions, etc.). Unguided reflection (e.g., open questions) was often perceived as being intrusive and threatening by both coaches and staff.

An important contextual variable that facilitated the success of reflection was the background of coaches themselves possessing relevant practice-based knowledge. This made reflection specific, which contributed to this process being a valuable learning tool. Another important condition was the timing of reflection, which was related to the creation of a safe

environment and bonds of trust. In addition, reflection in the absence of safety and trust appeared to raise the threshold regarding engagement and led coaches to being less involved and less responsive during sessions.

Under suitable conditions, discussion and reflection appeared to raise awareness and deepen the understanding of health promotion practices and coaching. Moreover, discussion and reflection seemed to provide coaches with the confidence and skills needed to communicate with at-risk youth regarding health promotion topics (e.g., about the benefits of regular physical activity, healthy eating, etc.) and to implement new health promotion actions in practice (e.g., providing healthy snacks during sport activities). Discussion and reflection also led staff to look beyond their expectations of the abilities of SFD coaches. Openness was created by discussing values and goals, and the staff expressed their urge to incorporate deepened reflection in their current practice to provide a facilitative context to strengthen the abilities of coaches to reach these goals.

*"(. . .) also with the team evaluation, I noticed that it went more in depth, that it doesn't just refer to 'it's good'. (. . .) it's not just about numbers and reach but also about the relationship with those guys* [youth]." (Staff member 1)

*"They* [the coaches] *were also thinking about who they were going to put together in a group and who not. And now thinking about it, there was also a coach who said 'hmm, that game in that neighborhood, we're not going to do that because of what we saw last week, that if we play that game in that neighborhood then we are actually making everyone feel uneasy."* (Staff member 2)

## Discussion

In the present study, we explored the feasibility of a training program (also referred to as an intervention) developed to increase the health awareness and skills among SFD coaches to enable them to become viable deliverers of health prevention messages to at-risk youth. We aimed to clarify how the training worked (mechanisms of impact) and which elements external to the training within the context of SFD led to its supposed effects (context factors), by conducting a process evaluation through a realist lens. A pre-developed program theory was used as the foundation for this study [34]. This theory has been further tested and refined in other SFD contexts [35]. It represents the effects of SFD programs on health outcomes in at-risk populations through mechanisms of experiential learning among participants and incremental responsibility taking and reflexivity. The mechanisms described in this program theory were successfully transferred provided that participants felt safe, were stimulated to reflect about their behavior, and were enabled to be agentic. This realist-informed process evaluation sought to refine a specific key element within the theory, i.e., the role of the coach. Given the need for specialized training of SFD coaches for the installation of the above effects and to efficiently spread health promotion messages, we explored how and when training programs with this purpose would be effective for coaches.

Overall, our findings showed that the training was suitable and well accepted by coaches who expressed positive responses to the training, i.e., it increased their awareness of health and their actions as role models for at-risk youth. This overall finding has great importance. One of the key change mechanisms of improved health among at-risk youth using sports as a lever is their observation and learning through what is respected by others and in particular with regards to what the coaches are doing [34, 35]. This process of 'vicarious learning' is most profoundly described in the Social Cognitive Theory (SCT) of Bandura [39].

Based on SCT, several pathways of learning could be presumed. The first is the most direct pathway: coaches may give information and instructions on how to change behavior. For example, a coach may explain the negative effects of smoking on overall health. The second pathway involves the health behavior of the coaches, which may motivate youth to behave in a similar way because of the benefits they perceive or hear the coaches express. For example, a coach expressing feeling fitter after having quit smoking may motivate youth to also quit smoking. Thirdly, coaches may serve as a social prompt for CYP to perform healthy behavior amongst different alternatives. For example, a coach may set an example by drinking water and eating fruit during sport activities. Despite the pathways differing, they are all deemed important. While the first two pathways may require more intentional, rational thinking (i.e., by gaining knowledge and changing attitudes), the latter pathway is likely to occur in a more automatic, unintentional manner. Within the Elaboration Likelihood Model of attitude change of Petty and Cacioppo [40], this difference is referred to as the central and peripheral route. The central route consists of thoughtful consideration of the arguments (ideas or content) of the message. The peripheral route occurs when the listener decides whether to agree with the message based on other cues, such as the bond or perceived power of the person delivering the message. Research suggests, albeit mostly in college students, that a few variables may influence the extent to which people are more likely to be convinced by contextual cues rather than by the message itself. Examples include not being motivated by the content of the message (e.g., having no interest or seeing no benefit), not being able to think about the message (e.g., because of being distracted by other things), or having to think about a message that is difficult to comprehend [41, 42]. Although it has not yet been investigated, at-risk youth are likely to be more sensitive to health messages delivered by coaches because of their status, trustworthiness, and perceived power, than through the content of the message itself. Therefore, the social prompting of health messages may seem to be an important route of transfer between coaches and at-risk youth, and is thus an assumption worthy of further investigation.

It is important to note here that, based on our findings, the training did not easily facilitate changes in the behavior of coaches, which is an important prerequisite for social prompts to occur. Coaches did not report changes in setting personal health goals themselves nor increased efforts towards accomplishing health goals or adopting healthy behavior. Those who reported change (e.g., a few coaches mentioned changes in their dietary intake or smoking behavior) mentioned the diminishing of the training effects on their behavior after a while. These findings were consistent with the staff observations that stressed that the training did not lead to observable, long lasting changes in health behavior despite challenging the coaches to question their health status and risky health behavior. However, these findings were not entirely remarkable and were consistent with theoretical ideas on behavior change. Process theories on behavior change, for example, delineate behavior change as a time-consuming endeavor that proceeds through different 'stages' [43, 44]. These stages differ between theories, but there are also communalities among theories. For example, it is commonly theorized that individuals first proceed through a stage of awareness and knowledge building before they express an intention to change their behavior, and for the better. In parallel, the training may ultimately be effective in changing the awareness and knowledge of participants, but for it to have a lasting impact on the behavior of coaches, more actions may be needed. Here, we advocate more intense individualized guidance and feedback of coaches, which could not be obtained from a group-based format. In addition, the time frame may need to be lengthened, so that long-term follow-ups are possible, and the opportunities to transfer skills to real-life settings are increased.

Performing the process evaluation through a realist lens provided us with in-depth insights into the inner workings of the training, and the contextual boundaries that have to be set for

the mechanisms of impact to result in the preferred outcome. As such, we gained in-depth knowledge on the supposed successes or failures of training outcomes as perceived by coaches, staff, and other stakeholders. Beyond the 'classic' process evaluation, realist thinking furthers our insights regarding the hidden causal forces behind the observed patterns or changes in those patterns. In a realist evaluation, this is achieved through retroduction, i.e., by looking back at observed patterns to determine what processes were responsible for creating them [28]. Here, we used a realist-informed approach because we aimed to determine the context conditions needed to generate the training mechanisms that led to some of the observed training outcomes. The implementation of the training was, however, far from perfect, as it did not manage to reach all SFD coaches, and one SFD coach even discontinued the involvement in the program. The context of SFD can be challenging since most coaches come from vulnerable situations themselves, and who are searching for prospects of better working conditions and life satisfaction. Our data indeed highlighted that low levels of self-esteem, mental health problems that are often a result of precarious family situations, and educational issues in SFD coaches could significantly hamper intervention delivery and impact. Below, we elaborate in detail the theory-of-change underlying the training, regarding its inner mechanisms and how they are generated within the particular SFD setting investigated in this study.

The first finding relates to the type of environment that was created at the start of the training and continued throughout the program, i.e., the establishment of a safe and trustworthy environment for participation in the intervention and for the personal development of SFD coaches. Our study showed that a sense of safety reduced the doubts and insecurities of coaches, and produced an environment that was conducive to learning and which did not punish individuals for not participating within the training or when missing one or more training components. This idea of 'psychological safety' is consistent with other scholarly publications on the conditions that need to be met for people to be motivated to perform, i.e., to learn and participate in the training in this respect. Based on Self-Determination Theory (SDT) [45], these studies highlighted that the creation of a social context that motivates participants or athletes to the degree in which they feel autonomous and sufficiently free, positively affects their level of intrinsic motivation and enjoyment (e.g., [46]). Indeed, the friendly attitude and openness that both the SFD coaches and staff expressed towards intervention deliverers was related to their increased awareness of the need to discuss the topics and intrinsic motivation to proceed with the intervention. However, this appears to be a precarious situation. It requires amicable, supportive, and inclusive but also professional relations between SFD coaches and staff as well as with intervention deliverers. Excessive professionalism (e.g., requiring the completion of assignments, strictness regarding punctuality, etc.) may create the perception of an unsafe environment and hamper the confidence of coaches to learn.

A second finding relates to the promotion of involvement and engagement. The use of a variety of activities and fun activities, vivid material, and an approach aimed at developing critical self-reflection and self-development appeared to be crucial. This creation of a sense of critical self-reflection appeared to increase health awareness among coaches and support them in their acting as role models. With regards to the staff, reflection instilled a sense of urgency for more than just the development of sport technical skills and to prepare SFD coaches for the delivery of health prevention messages as well as to, ultimately, become well-equipped health professionals with increased chances of better work prospects outside the particular SFD context. There was also a recognition that this requires training and guidance from another perspective, i.e., one that takes into account continuous reflection and individual guidance. Ultimately, among both coaches and staff, it was noticeable that such an increased awareness set the foundation for self-efficacy to grow, and confidence was expressed that training would be successful in guiding at-risk youths towards personal health agency. However, this depends

on whether the involvement and engagement does not increase the personal vulnerabilities of participants, such as uncertainties or doubts if one is able to live up to being a role model, which are frequently present in coaches due to life history events or precarious living situations, or both. When vulnerability is increased, it hampers the learning through training and shifts in thinking or awareness of being a role model for at-risk youths.

A parallel can be drawn with the mentoring theory of Pawson [47], which illustrates that different mentoring processes can lead to optimal personal development. Most notable in this respect is the process of 'direction setting,' which closely resembles the promotion of involvement and engagement, as observed in our data. In this theory, direction setting is described as a cognitive component that entails the promotion of self-reflection via discussion of alternatives or the setting up of individual learning plans, and a reconsideration of values, loyalties, and ambitions. Also, according to this theory, the element of direction setting combined with the process of 'befriending or creating bonds of trust,' and 'coaching or the acquisition of skills,' leads to successful mentoring programs [47].

Befriending relates to another main working process in our data: *group dynamics and relations*. We found that meaningful bonds were developed and intensified, and that coaches and staff continued to build confidence to learn more. Especially, the creation of an informal context of interaction during the intervention (e.g., talking freely, being listened to, and group building activities) appeared to provide opportunities for coaches and staff to build and intensify meaningful and respectful relationships, which led to the SFD coaches having improved confidence and being able to observe that their actions (however small they were) were able to have an impact. To play a role in a group, to be part of a greater whole, and to be connected with others gives people the feeling they have the right to be. It increases self-confidence and motivation to change one's behavior in the long run. According to SDT [45], a sense of relatedness is indeed one of three sources, besides and in relation to competence and autonomy, through which motivation can thrive.

Successful mentoring also involves coaching or assistance in acquiring new skills [47]. This element is significantly related to the process of discussion and reflection. When the right conditions were in place, such as the use of the wealth of vivid experiences of coaches and the timing of reflection, this process appeared to raise awareness and deepen the understanding of health promotion practices and coaching. Moreover, discussion and reflection seemed to provide coaches with the confidence and skills needed to communicate health promotion topics to at-risk youth (e.g., regarding the benefits of regular physical activity, healthy eating, etc.) and to implement new health promotion actions (e.g., providing healthy snacks during sport activities). Ideally, we would have wanted to have observed coaches changing their own health behavior and live model lifestyles. As previously described, such changes would make them more effective deliverers of health prevention messages, and would increase the potential for behavior change among at-risk youth in the long term. In addition, contextual variables have to be present during the processes to produce these outcomes, e.g., having enough time, and to successfully create safe opportunities to discuss personal progress and goals. The latter element was not developed within the current training. More is needed to translate motivation into personal action plans and behavior change [48, 49], e.g., assisting individuals in setting realistic goals, planning towards these goals, and helping to conquer the obstacles that hinder them in achieving their goals (e.g., [50, 51]).

## Strengths and limitations

First, a case study approach was central to this study and allowed for an in-depth analysis of the theory-of-change underlying the roles of SFD coaches in using sport as a vehicle for

improving the health of at-risk youth. Several mechanisms were identified that were believed to affect the attitudes and behaviors of coaches, which further triggered health promotion practice in at-risk youth. We followed a realist theory perspective [28], assuming that interventions involve not one, but several theories or mechanisms that lead to its effects, under specific circumstances. The main processes observed appeared to be essential for the successful training of the coaches, however, precaution is warranted regarding the impacts of such training. These impacts may be limited by the presence of contextual variables, such as the level of (health) literacy, mental health, and the living conditions of coaches. As observed in the present study, context may also facilitate training among coaches and staff with already established bonds of trust. This study offers insights into measures to be taken when implementing and evaluating similar training programs elsewhere. Evidently, and not explored within this study, structural influential factors may also impact the success or failure of SFD for health. Examples include (local) policy decisions, or macroeconomic factors regarding housing or prevention budgets. Also, certain groups may be more prone to encounter negative peer pressure, or may live in or have encountered precarious family situations that negatively impact their agency to take control of their health and life. Recent theories have emphasized the influence of such environmental influential factors on the motivation of certain individuals to perform health behavior, and for behavior change [52]. Future research is needed to identify these factors within the context of SFD, as well as their interaction with more proximate context factors and individual agentic determinants.

Second, we used a systematic approach to design and conduct our process evaluation, using the MRC guide [29]. Hearing the views of multiple stakeholders and the use of different data collection methods (observations, interviews, focus group, self-reports) furthered our understanding of the complexity of the inner workings of the SFD intervention. Some of the interviews with SFD coaches were difficult, wherein some struggled to elaborate on questions regarding supposed mechanisms, context factors, and outcomes. We used triangulation to solve this issue, combining multiple observers and methods. We are, therefore, confident that our results are valid and reliable.

Third, the set-up of this process evaluation was developed in close collaboration with staff and other local stakeholders and is, thus, a strength. Spaaij et al. [53] stated that in situations where there is a high degree of participation, mutual learning is established and obstacles of cultural boundaries are lifted. However, measurement instruments, such as observation and interview guides, were developed independent from stakeholders. A crucial factor in this situation was time. Nevertheless, it would be worthwhile to co-construct evaluation timeframes and measures, leaving the control of monitoring and evaluation to participants, and thus increasing the chance of successful embedding of these methods in practice.

Fourth, two researchers (KVDV, EL), who observed the meetings and interviewed the participants, also functioned as tutors. We considered this to be a strength because a meaningful relationship had been developed with the SFD coaches, thus contributing to the vividness of experiences told during the interviews. However, this engagement in quality improvement activities may also compromise the external validity of the evaluation. We tried to solve this issue as much as possible, with a third researcher acting as a passive observer who did not intervene and did not feed findings back to the other researchers, thus minimizing the effects of this possible limitation.

Fifth, we conducted a process evaluation and were interested in the feasibility and theory-of-change underlying the feasibility, i.e., processes at work that impacted training outcomes, under specific circumstances. We used a qualitative research design to gain in-depth knowledge regarding the feasibility and, in particular, the experiences surrounding the training, underlying processes, and the complex links with outcomes and associated contextual

variables. Quantitative or mixed-method designs can be used in future studies to test these pre-hypothesized pathways of impact, and contextual moderators. In future studies, data should be collected at multiple time points to capture the effects of the training over time.

## Conclusion

A realist-informed process evaluation was deemed the most suitable approach to determine the feasibility of training to increase the awareness and ability of coaches to transfer health prevention messages to at-risk youth. We used the MRC guidelines as a framework for our analysis. We were also interested in clarifying the underlying mechanisms of the impact and context factors that led to the supposed training effects. A case study was undertaken allowing the collection of views from multiple stakeholders and from a variety of sources. A safe learning environment seemed to contribute to a 'right mind-set,' which facilitated the learning process among peers and tutors with whom the coaches felt connected. An increased health awareness and sense of responsibility to act as a role model for at-risk youth were among the main outcomes, and was reached through an increase in self-confidence, and an improved sense of critical self-reflection and self-development. These outcomes were triggered through the above-described processes under certain conditions. In addition, several variables, such as a precarious life history or living conditions, mental health issues, or low educational skills, may hamper the processes and outcomes. This study offers valuable insights into the processes and appropriate circumstances of SFD training that may prepare coaches to effectively deliver health prevention messages. This may inform intervention developers and policy makers to make more sensitive and suitable choices in the setting up and implementation of health prevention programs through sports using coaches as deliverers of those messages. Additionally, programs such as the one used in the present study are very unlikely to directly impact health behavior, but rather set the stage for further individualized actions to be undertaken. Therefore, programs should be assessed and monitored using the intermediate outcomes most at stake. This includes motivation, and even more proximate indicators, such as awareness, self-efficacy, sense of self-reflection, and a sense of responsibility as good candidates.

## Acknowledgments

The authors thank the staff and SFD coaches of 'Buurtsport Brugge' for their collaboration with the study implementation and data collection, and April Ward and Ellen Bruffaerts for their assistance in translating the quotes. In particular, we are very grateful to Dr. Kevin Harris for his critical appraisal of a pre-final version of the manuscript.

## Author Contributions

**Conceptualization:** Emelien Lauwerier, Karen Van der Veken, Sara Willems.

**Data curation:** Emelien Lauwerier, Karen Van der Veken.

**Formal analysis:** Emelien Lauwerier, Esther Van Poel, Kaatje Van Roy.

**Funding acquisition:** Sara Willems.

**Investigation:** Emelien Lauwerier, Esther Van Poel, Karen Van der Veken.

**Methodology:** Emelien Lauwerier, Karen Van der Veken, Sara Willems.

**Project administration:** Emelien Lauwerier, Karen Van der Veken.

**Resources:** Emelien Lauwerier, Karen Van der Veken, Sara Willems.

**Software:** Esther Van Poel.

**Supervision:** Sara Willems.

**Validation:** Emelien Lauwerier.

**Visualization:** Emelien Lauwerier.

**Writing – original draft:** Emelien Lauwerier.

**Writing – review & editing:** Emelien Lauwerier, Esther Van Poel, Karen Van der Veken, Kaatje Van Roy, Sara Willems.

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
