## [Decision Letter · Decision Letter 0]

10 Feb 2020

PONE-D-20-01877

A realist process evaluation of a sport-for-development intervention to improve the health of at-risk youth

PLOS ONE

Dear Prof.dr. Lauwerier,

Thank you for submitting your manuscript to PLOS ONE. After careful consideration, we feel that it has merit but does not fully meet PLOS ONE’s publication criteria as it currently stands. Therefore, we invite you to submit a revised version of the manuscript that addresses the points raised during the review process.

please address all the issue raised by the reviewers before resubmitting your manuscript.

We would appreciate receiving your revised manuscript by Mar 26 2020 11:59PM. To enhance the reproducibility of your results, we recommend that if applicable you deposit your laboratory protocols in protocols.io, where a protocol can be assigned its own identifier (DOI) such that it can be cited independently in the future. For instructions see: http://journals.plos.org/plosone/s/submission-guidelines#loc-laboratory-protocols

We look forward to receiving your revised manuscript.

Kind regards,

Elena Cavarretta, M.D., Ph.D.

Academic Editor

PLOS ONE

Journal Requirements:

1. Please include additional information regarding the survey or questionnaire used in the study and ensure that you have provided sufficient details that others could replicate the analyses. For instance, if you developed a questionnaire as part of this study and it is not under a copyright more restrictive than CC-BY, please include a copy, in both the original language and English, as Supporting Information. Moreover, please include more details on how the questionnaire was pre-tested, and whether it was validated.

2. When reporting the results of qualitative research, we suggest consulting the COREQ guidelines: http://intqhc.oxfordjournals.org/content/19/6/349. In this case, please consider including more information on the number of interviewers, their training and characteristics.

3. Please ensure that all statements made (for example, the ones at lines 86-87 of the Introduction) are supported by relevant citations .

4. Please include your tables as part of your main manuscript and remove the individual files. Please note that supplementary tables (should remain/ be uploaded) as separate "supporting information" files

Reviewers' comments:

Reviewer's Responses to Questions

**Comments to the Author**

1. Is the manuscript technically sound, and do the data support the conclusions?

Reviewer #1: Partly

Reviewer #2: Partly

2. Has the statistical analysis been performed appropriately and rigorously? 

Reviewer #1: Yes

Reviewer #2: Yes

3. Have the authors made all data underlying the findings in their manuscript fully available?

Reviewer #1: Yes

Reviewer #2: No

4. Is the manuscript presented in an intelligible fashion and written in standard English?

Reviewer #1: Yes

Reviewer #2: No

5. Review Comments to the Author

Reviewer #1: The authors conducted a process evaluation of an intervention for coaches that work in a Sport-for-development program (SFD). They touch upon a number of interesting aspects in an in-depth case study, describing how various mechanisms play a role in training these coaches for their future work with at-risk youth. As the coaches themselves have a vulnerable background, this offers an interesting view on how interventions like this are perceived and impact the coaches’ behaviour.

Having said that the data is interesting, there are a number of points for improvement. First, I question whether a true realist evaluation has been conducted. It seems that the material does not offer insights on context, mechanisms and outcomes, and how these are linked. Secondly, I would like to urge the authors to offer a more critical perspective on the process, as on several occasions I get the impression that the delivery of the intervention was not optimal and that the impact of the intervention on actual behaviour may have been limited. Thirdly, the authors seem to switch between talking about the mechanisms underlying the training for coaches and the mechanisms of the SFD program that the coaches are implementing for youth. This is very confusing at times, and it leads to conclusions being drawn about the impact of the SFD program that in my opinion cannot be drawn.

Several major and minor issues are elaborated upon in the attached document. I hope my comments are received in the same way in which they are intended – as positive and constructive encouragement.

Reviewer #2: This study about an intervention for SFD coaches is very interesting because it is a process evaluation, and the execution of multiple methods. The intervention has the goal to inform and train SFD coaches to coach CYP regarding their health related behaviour. The intervention focuses on knowledge acquisition for unhealthy behavior and mechanisms to promote health, and coaching skills. Coaches appreciated evaluation and reflection most and the fact that they were confirmed in their qualities, and became aware of their role and position.

Method: The chosen method is very suitable and extensive to evaluate the intervention. It brings above the experiences which are necessary to implement or adjust the intervention. However, with the description of the method tells us that this study will reveal how the intervention was experienced by the coaches. The title is misleading due to the focus on the improvement of health of youth and would be better with a focus on coaches, who have to affect the health related behavior of youth.

Data: The data supports the conclusion partly because the conclusion is too general and not critical enough.

The 5 points were clear and helpful for other interventions. Point 4 triggered because I think it was about awareness in stead of setting personal goals.

The intervention was good but not fully developed. This study gives the opportunity to develop the intervention further. However, a critical analyse is therefore necessary. As a reader, I think the coaches appreciate the reflection, discussion and theory about awareness and knowledge about promoting health. They already have the knowledge about unhealthy behavior, and due to the focus of the first sessions on this theme, presence can be lowered. It would be helpful if new ideas would be presented to create more presence. And if individual sessions are desired amd what should be the focus of these sessions.

Analysis: Analysis were suitable and good.

Availability of data: This study obtained a lot of useful information. The researchers declare that all data is available. However it would be useful if we get an insight in the result of the thematic analyse, questionaire results. It is not clear if these results will be available in an appendix.

English: There were a few minor elements such as a missing space in the abstract.

A few questions which can be described in the article:

- The introduction focuses on access to services, but isn't it about health literacy?

- Only male participants and from Bruges. Do you think the intervention is suitable for women and somewhere else?

- How did you decide that fidelity can be marked as good? The same question arised when you state that the intervention is suitable and accepted.

- Is the timeframe large enough, or should it be lengthened so the staff can follow coaches their development and train them in real life settings?

- Which staf member of coach stated the quotations? It would be helpful if everyone has a number, so it is visisble that the quotations do not stem from 1 or 2 persons.

6. PLOS authors have the option to publish the peer review history of their article (what does this mean?). If published, this will include your full peer review and any attached files.

Reviewer #1: No

Reviewer #2: No

---

## [Author Response · Author response to Decision Letter 0]

2 Jun 2020

Comments to the Author

1. Is the manuscript technically sound, and do the data support the conclusions?

Reviewer #1: Partly

Reviewer #2: Partly

After major revision and final editing of our paper, the technical soundness of the research should be undoubtedly and unvaguely be presented to readers. 

2. Has the statistical analysis been performed appropriately and rigorously? 

Reviewer #1: Yes

Reviewer #2: Yes

3. Have the authors made all data underlying the findings in their manuscript fully available?

Reviewer #1: Yes

Reviewer #2: No

We carefully read the PLOS Data policy, and do confirm that the data (quotes) are provided within the paper and reflect all data from which interpretations and conclusions were made.

4. Is the manuscript presented in an intelligible fashion and written in standard English?

Reviewer #1: Yes

Reviewer #2: No

Based upon this and other comments, and due to the fact that a major revision was performed and significant changes were made compared to the original paper, we decided to use professional language editing. We are confident that the final paper now entirely complies with all requirements of the journal and is written in an intelligible fashion and in standard English. 

5. Review Comments to the Author

Reviewer #1: The authors conducted a process evaluation of an intervention for coaches that work in a Sport-for-development program (SFD). They touch upon a number of interesting aspects in an in-depth case study, describing how various mechanisms play a role in training these coaches for their future work with at-risk youth. As the coaches themselves have a vulnerable background, this offers an interesting view on how interventions like this are perceived and impact the coaches’ behaviour.

This is an excellent summary of the content and aims of our study, and we would like to thank the reviewer for this kind appreciation. 

Having said that the data is interesting, there are a number of points for improvement. First, I question whether a true realist evaluation has been conducted. It seems that the material does not offer insights on context, mechanisms and outcomes, and how these are linked. Secondly, I would like to urge the authors to offer a more critical perspective on the process, as on several occasions I get the impression that the delivery of the intervention was not optimal and that the impact of the intervention on actual behaviour may have been limited. Thirdly, the authors seem to switch between talking about the mechanisms underlying the training for coaches and the mechanisms of the SFD program that the coaches are implementing for youth. This is very confusing at times, and it leads to conclusions being drawn about the impact of the SFD program that in my opinion cannot be drawn.

We would like to thank the reviewer for having raised these issues. We chose to answer these three main concerns hereunder, but do also refer to our specific amendments based on the list of comments within the word file provided by the reviewer (see below). 

First, the reviewer shares the concern of the study being a true realist evaluation. We agree that our approach is not a realist evaluation study per se, but rather a realist-informed approach. We do acknowledge the paper missed critical information needed to understand our approach. For example, in the introduction section, there is more information needed on what constitutes a process evaluation, and what makes our process evaluation to be realist-informed. Our aim was to explore feasibility of the training for coaches, and in particular we wanted to test the training’s underlying theory about “what might cause change” within the specific context of sport-for-development. Therefore, we collected data not just about the implementation of the training, but also about those specific aspects of the SFD context that might impact the training outcomes, and about the specific mechanisms that might create change. In analyzing our data, we used both inductive and deductive reasoning. This way of analyzing is typical to a realist approach, and referred to as retroduction. Retroduction is the identification of hidden causal forces that lie behind observable, identified patterns, or changes in those patterns (Gilmore et al., 2019). The underlying social and psychological drivers (being the mechanism and context elements underlying the training) that we sought to explore were based on the insights of an initial programme theory developed earlier to this study. In this respect, it is informative that the current study is part of a four-year (2016-2019) research project – CATCH (Community Sports for AT-risk youth: innovative strategies for promoting personal development, health, and social CoHesion) – aimed at the exploration of how and when low-treshold sport practices have their effect in promoting social inclusion. In the first phase of this project, an initial program theory was developed based on a multiple case study and insights from literature review (Van der Veken et al., 2020a). This program theory is now being tested and refined, and several research articles are already written out (Van der Veken et al., 2020b) or are forthcoming. The current study is also best conceived of as a refinement of a part of the initial program theory. Within the current study, we focus on the in-depth exploration of one of the mechanisms of the program theory, namely how do coaches of low threshold sport practices ought to be trained to provide a health-promoting environment for at-risk youth, and under which circumstances does such a training intervention work or not. Admittedly, the stages of realist evaluation that informed this process evaluation were not visible in the paper. We originally chose to leave this information out because we assumed it not to be crucial to understand the current study aims and findings. We now choose differently. We added some brief information regarding the program theory as an output of previous realist research stages reported in other publications in the methods section. Also, we added information regarding our approach being realist-informed and largely inductive to the methods section. Overall and throughout the paper (but within the introduction in particular), we clarified the importance of using a realist lens while performing a process evaluation (for specific amendments, see comments below).

A second major concern relates to holding our results against a critical lens. The reviewer is right that intervention delivery was not ideal; a finding that we discussed in the paper in relation to what could have produced these results. Possibly, this “context-setting” could have given the impression that we perceived the intervention itself as successful, while we mainly wanted to understand the effect in light of the processes being put in place within given circumstances. We do not want to give a wrong impression, and therefore carefully revised the manuscript to this aim, in particular the discussion section of the paper (for specific amendments, see comments below). 

A third major concern – which we also do acknowledge – was related to the possible confusion of us writing about the processes underlying the training (leading to better health among at-risk youth) and the processes of the training itself (leading to improved health-promoting behaviour among other outcomes in coaches). We thoroughly revised the paper, and resolved possible confusion. By also adding the information on the theory formulation and initial program theory, we believe the aim of the current paper is more clear, in a sense that we intended to unravel the underlying processes and contextual elements influencing the possible impact of an intervention targeted at coaches’ behaviour to deliver health prevention messages within a SFD environment (for specific amendments, see below).

Several major and minor issues are elaborated upon in the attached document. I hope my comments are received in the same way in which they are intended – as positive and constructive encouragement.

We certainly are grateful for the time the reviewer has taken to go in depth through our submission, and we are convinced that all points raised let us to improve the paper in a very significant way.

Major issues

1. The introduction describes how young people from low SES groups run a higher risk of health issues. In the second paragraph, the authors focus on inequity in access to health services as an important cause in these health differences and how SFD programs can play an important role in prevention of health issues. Although SFD programs can indeed offer a setting in which health promotion efforts can be implemented, whether this also offers potential for reducing health inequities is questionable. I would like to urge the authors to be a bit more careful in and critical towards the potential of SFD programs to reach this. Perhaps it would help to be more specific on what they are investigating: a specific training for SFD coaches to implement health-behaviour messages in a sport program (like it is stated on line 105).

Indeed, the reviewer is right in stating that we should write with more caution regarding the value of SFD programs reducing health inequities. Of course, there is a multitude of initiatives needed in various domains for health inequity to be reduced. We thoroughly revised the paper for possible over-enthusiast writing, and adapted several parts, mainly within the abstract and introduction. 

“Initiatives that improve the access to and adoption of health promotion messages are important undertakings, e.g., sport. Sport-for-development (SFD) programs are seen as valuable delivery tools, in which coaches are used as change agents to increase health awareness and behavior among at-risk youth.” (page 2, lines 20-23)

“Inequities in long-term health can be reduced not only by improved access to health promotion and prevention services but also by the tailoring of these services to the circumstances of CYP, among other initiatives [8]. Sport-for-development (SFD) may provide a setting through which health promotion and prevention messages can more easily reach and be adopted by hard-to-reach populations” (pages 3-4, lines 62-66)

Next, we noticed thanks to the comment of the reviewer that possible confusion could arise regarding the aim of the study. As the reviewer correctly summarizes, our aim was to evaluate the feasibility of a training for coaches to deliver health prevention messages in a sport-for-development setting. In the end, the aim is of course that their learning would transfer into more awareness and health behaviour among at-risk youth, but this was not aimed for in the study nor investigated. We revised all sections where possible confusion could arise, and most notable within the abstract, introduction and discussion sections. 

“The delivery of such messages requires specific knowledge and skills that can be attained through training; however, the effectiveness of such training requires assessment. In this study, we evaluated the feasibility of such a training program for SFD coaches using process evaluation from a realist perspective, and views from multiple stakeholders, among other sources. We also clarified the inner workings of the training and investigated how context shaped the training outcomes” (page 2, lines 23-29)

“The aim of this study, was to gather data from multiple stakeholders within this specific context, including SFD coaches, staff, and local policy makers, to enable an in-depth analysis of the feasibility of the training. The study objectives were to determine how to effectively train coaches to ensure their viability as deliverers of health prevention messages to at-risk youth populations, and clarify which elements external to the training and specific to SFD programs can lead to the successes or failures of such training” (pages 5-6, lines 111-117). 

“In the present study, we explored the feasibility of a training program (also referred to as an intervention) developed to increase the health awareness and skills among SFD coaches to enable them to become viable deliverers of health prevention messages to at-risk youth. We aimed to clarify how the training worked (mechanisms of impact) and which elements external to the training within the context of SFD led to its supposed effects (context factors), by conducting a process evaluation through a realist lens” (page 26, lines 586-591)

“A realist-informed process evaluation was deemed the most suitable approach to determine the feasibility of training to increase the awareness and ability of coaches to transfer health prevention messages to at-risk youth. We used the MRC guidelines as a framework for our analysis. We were also interested in clarifying the underlying mechanisms of the impact and context factors that led to the supposed training effects.” (page 35, lines 813-817) 

2. The study focuses on the evaluation of an intervention for SFD coaches and what mechanisms underly this training that may impact the coaches’ behaviour. So in my opinion the context is the intervention for coaches and the outcome is the coaches’ behaviour in the SFD program that is given to at-risk youth. However, while reading the introduction, I was confused to see that the authors also talk about the outcome of the SFD program on the health of at-risk youth, which is not the object of study. Alike, in the results and discussion the authors seem to talk about mechanisms underlying the SFD program for at-risk youth, and not the training for the coaches. Perhaps this has to do with the fact that the authors use a lot of anaphora in the text, which makes it difficult to identify what they are referring to. For example on line 563-566, the authors write that “they conducted a sport-for-development intervention targeting coaches with the aim of improving the health of at-risk youth. We were mainly interesting in how the program works...”. Does ‘the program’ relate to the SFD program for at-risk youth or the training for coaches. It would help to be more specific on this and perhaps even omit references to the SFD program for youth from the manuscript. Similarly, ‘impacting the health of at-risk youth’ is not object of study, so perhaps could be deleted in the text to avoid confusion.

Yes, as already mentioned in response to one of the reviewer’s main concerns, we fully acknowledge this possible confusion. The reviewer is perfectly right in stating that the aim is about ‘testing’ the coach training on coaches’ behavior, and not about evaluating the impact of the training on the health of at-risk youth. We carefully revised the paper, and most notably the introduction, results and discussion sections to which the reviewer refers in particular. The amendments are that significant that we choose not to refer to specific sections below. Of course, all changes are indicated in-text.

3. The authors talk about a realist evaluation, analysing the ‘theory-of-change’, describing context-mechanisms-outcomes, but in reality they are not actually doing a realist evaluation, but rather a ‘normal’ process evaluation, which is also fine given the aim of the study. The training is developed based on a theory-of-change, which is then evaluated by taking notes during the training and interviews/focus group. This does not allow for really falsifying the mechanisms and so I would refrain from making statements about identifying which mechanisms were important or what outcomes were reached. Perhaps it is a suggestion to reflect through the interviews on how these mechanisms were perceived by the coaches. This will be better supported by the data. In this respect, for example on line 462-465 the authors talk about ‘barriers that moderate intervention impact’, but they do not analyse COM configurations. 

As already mentioned above in responding to one of the main concerns of the reviewer, our main aim was indeed to perform a process evaluation, and intentionally one that was realist-informed. 

We made the following changes to be more clear about the why and how of the methodology used: 

1) we optimized our introduction section describing why performing a realist-informed process evaluation is of choice in light of our study aims; 

2) we added some brief information about the program theory that was developed prior to the current study and that informed our analysis; 

3) we added some more information about the CATCH project of which the current study was part of, to make clear the sequence of steps that led to the current study, being a refinement of a specific element within a pre-defined program theory; and 

4) we added some brief information regarding our approach to be realist-informed (and not a realist evaluation per se), as well as specified our data-analysis approach within the methods section. 

“To achieve this, we conducted a process evaluation built on realist theory [28]. Realist theory recognizes that many variables operate at different levels, and which account for differences in program effects. It is imperative to acknowledge that interventions do not necessarily work for everyone, because of differences among people and the contexts that they are embedded in. It is, therefore, vital to clarify which elements influence the effectiveness of programs (i.e., mechanisms of impact) and which external variables (i.e., context elements) may hamper or facilitate their impacts [29].” (page 5, lines 95-102)

“The current study forms part of a four-year (2016–2019) research project, named CATCH – Community sports for AT-risk youth: innovative strategies for promoting personal development, health, and social CoHesion – which is aimed at exploring ‘why, how, for whom, and under which circumstances’ community sport activities enhance the personal development, health, and social cohesion of at-risk youth. The outputs of previous studies resulted in the development of a program theory and its further refinement by clarifying why, how, and when community sport can promote health, in terms of the mechanisms and context factors of improved health outcomes of at-risk youth. These results are described elsewhere [34,35], and a training program was developed for community coaches based on the insights of the program theory [31]. A brief overview of this training program is needed to enable the interpretation of the current study, and it is, therefore, described below.” (page 7, lines 140-150)

“In particular, we used a realist-informed process evaluation (and not a realist evaluation per se) to explore this ‘theory-of-change.’ Realist evaluations consist of a set of methods and are characterized by an iterative set of stages that involve developing, testing, and refining a theory. Our analysis approach was more inductive, open-ended, and sought to document how the training resources promoted the changes among participants that led to the observed outcomes, under specific circumstances [36]. We initially assessed the empirical data, and then tried to identify hidden processes and develop ideas using a realist data analysis process called ‘retroduction’ [37] to clarify what aspects of the training worked for SFD coaches, why the training was effective, and under which circumstances the outcomes occurred. Our identification of hidden processes was informed by a pre-developed program theory on the mechanisms and influencing context factors of improved health among at-risk youth [34]. The current study is best conceived as an in-depth refinement of one of the key-influencing factors within this program theory, being the application of motivational coaching and the installation of positive group dynamics by coaches to increase the health outcomes of at-risk youth. Central mechanisms of impact are experiential learning, incremental responsibility-taking, and reflexivity that may, under the right circumstances (e.g., participants feeling safe around others), beneficially influence the health outcomes. These processes and some of the contextual variables may be transferable to the coach training situation regarding health promotion within this same setting of community sports because the local coaches appear to have similar at-risk backgrounds compared to the youth themselves. Therefore, the mechanisms and contextual variables within the program theory provided a good starting point from which interpretations and hidden influential factors could be inferred from the data in the present study. A visual scheme depicting a ‘translation’ of mechanisms of impact, contextual variables, and outcomes is presented in Fig 1.” (pages 8-9, lines 187-210) 

Also, we revised the results and discussion sections and adapted wordings wherever we felt we had to stay close to the study aims and methodology. These changes are not provided below, but are all indicated throughout the paper, and within the results section in particular.

4. The results section is set up around the mechanisms that were part of the developed intervention. They are called ‘main working elements’ at this point, which adds to the confusion of the reader of what is actually being discussed here. It seems as the authors are discussing how the activities in the coaches’ training are perceived by the coaches themselves and whether the activities actually reflected the mechanisms that were intended by the researchers/developers of the training. I would focus on that, and refrain from presenting data on other aspects, for example how this relates to the SFD program that is implemented for youth and how this impacts these youths’ health. 

We feel there is raised some confusion due to the wordings within our results section. The reviewer indeed is right that we wanted to discuss how the training, and in particular its supposed mechanisms of change and contextual variables that may impact outcome and implementation, was perceived by interviewees (not only coaches) and through observations. We thoroughly revised this section and resolved any possible confusion. All changes are indicated in-text. 

5. The discussion consists of several sections. The summary (1st paragraph) is not really accurately portraying the results in my opinion. See also my previous remarks about this not being a realist evaluation. The reference to mechanisms of impact and context factors is out of place. They also talk about ‘intervention elements’ (line 570) which is a new term used. The summary of mechanisms is not accurate because some were shown in the results section to be not relevant. 

The reviewer all rightfully notices this. We re-wrote this first paragraph significantly and it now represents a short summary of the study aims and approach. In describing this, we were also more precise on our approach being a realist-informed process evaluation (see also above). A description and in-depth discussion of mechanisms of impact perceived within the training is now described further on within the discussion. 

“In the present study, we explored the feasibility of a training program (also referred to as an intervention) developed to increase the health awareness and skills among SFD coaches to enable them to become viable deliverers of health prevention messages to at-risk youth. We aimed to clarify how the training worked (mechanisms of impact) and which elements external to the training within the context of SFD led to its supposed effects (context factors), by conducting a process evaluation through a realist lens. A pre-developed program theory was used as the foundation for this study [34]. This theory has been further tested and refined in other SFD contexts [35]. It represents the effects of SFD programs on health outcomes in at-risk populations through mechanisms of experiential learning among participants and incremental responsibility taking and reflexivity. The mechanisms described in this program theory were successfully transferred provided that participants felt safe, were stimulated to reflect about their behavior, and were enabled to be agentic. This realist-informed process evaluation sought to refine a specific key element within the theory, i.e., the role of the coach. Given the need for specialized training of SFD coaches for the installation of the above effects and to efficiently spread health promotion messages, we explored how and when training programs with this purpose would be effective for coaches.” (pages 26-27, lines 586-601)

6. The discussion goes on to summarise the different mechanisms as laid out in the change theory, but it does not seem to add something new. These paragraphs could be deleted or become part of the results to explain the mechanisms more in-depth. In addition, on several occasions I felt that the authors draw conclusions that are not supported by the results, or at least they do not seem to directly refer to results that were discussed in the results section. New materials and data are offered here (for example. Lines 588-593: internal motivation was raised OR lines 607-608: about mutual trust). Finally, the authors could be more critical towards the results, because there seems to be drop-out and low dose and fidelity. 

Indeed, while it was of course not our intention, we acknowledge that the discussion section missed some focus at times. In our opinion, the fact that there was some confusion about study aims and approach may have added to the critique. We significantly revised our discussion section. From a structural point of view, we now chose to start with discussing the overall feasibility of the training, and a critical appraisal of drop-out and poor delivery at times. Next, we present the reader with an in-depth view on the mechanisms of change and contextual variables that impact implementation and training outcomes. These findings are imperative to understanding successes of the training, but also its weaknesses. We used the concept of ‘internal motivation’ as this is explicitly stated within the pre-developed program theory that served as a starting point from which to derive hidden mechanisms out of our data. But we do admit now that this concept is not easily understood within the current study, mainly because a detailed description of this program theory is out of the scope of this study. We omitted this and other concepts that do not necessarily have to be mentioned within the current study. We do not indicate all changes below. We are willing to provide these to the reviewer, but kindly refer to the revised paragraphs within the text (in particular all paragraphs preceding the section ‘strengths and limitations’) (see pages 26-33). 

7. The conclusion starting at line 738 is too strongly formulated and is not supported by the results. Confusion in use of wording (e.g. employees?, perceived intervention effects etc.). Disparities in dose and fidelity were a strength? I do not agree. A key element was personal health goals, but the authors have found that this was not a working element. 

We thank the reviewer and agree that the conclusion could be improved. Again, we notice some confusion due to our wordings regarding our study aims, approach, and how we perceived our results, and we apologize for this. We cleared all possible confusion, and re-wrote it entirely. 

“A realist-informed process evaluation was deemed the most suitable approach to determine the feasibility of training to increase the awareness and ability of coaches to transfer health prevention messages to at-risk youth. We used the MRC guidelines as a framework for our analysis. We were also interested in clarifying the underlying mechanisms of the impact and context factors that led to the supposed training effects. A case study was undertaken allowing the collection of views from multiple stakeholders and from a variety of sources. A safe learning environment seemed to contribute to a ‘right mind-set,’ which facilitated the learning process among peers and tutors with whom the coaches felt connected. An increased health awareness and sense of responsibility to act as a role model for at-risk youth were among the main outcomes, and was reached through an increase in self-confidence, and an improved sense of critical self-reflection and self-development. These outcomes were triggered through the above-described processes under certain conditions. In addition, several variables, such as a precarious life history or living conditions, mental health issues, or low educational skills, may hamper the processes and outcomes. This study offers valuable insights into the processes and appropriate circumstances of SFD training that may prepare coaches to effectively deliver health prevention messages. This may inform intervention developers and policy makers to make more sensitive and suitable choices in the setting up and implementation of health prevention programs through sports using coaches as deliverers of those messages. Additionally, programs such as the one used in the present study are very unlikely to directly impact health behavior, but rather set the stage for further individualized actions to be undertaken. Therefore, programs should be assessed and monitored using the intermediate outcomes most at stake. This includes motivation, and even more proximate indicators, such as awareness, self-efficacy, sense of self-reflection, and a sense of responsibility as good candidates” (pages 35-36, lines 813-836)

Minor issues

1. Abstract: line 35. ‘In their approach…’ This sentence is a big vague and without context. In addition, I am not able to trace this conclusion back to any of the results that I have read. 

The abstract has been entirely re-written in line with the amendments made in the paper. The sentence noticed by the reviewer is now left out.

2. Methods: I would like to see more information about how the intervention was developed. Based on literature, and if so, what literature? 

Although we understand the request of the reviewer, the development of the training is out of the scope of the current paper. Its description was worth a paper on its own, and is published elsewhere. We now refer to this publication in the current paper. 

“These results are described elsewhere [34,35], and a training program was developed for community coaches based on the insights of the program theory [31]. A brief overview of this training program is needed to enable the interpretation of the current study, and it is, therefore, described below.” (page 7, lines 147-150)

3. Methods: line 134. The mechanisms to promote health of CYP are discussed here, but how are they relevant for mechanisms for the coaches’ training? Why is the results section structured around the mechanisms for health promotion for CYP? (see also point 2, major issues)

This is indeed an excellent point, however, we can easily explain it and confusion can be resolved. We kindly refer to our answer to point 2 above. Below, we copy-paste the specific amendments made in the methods section. We omitted reference to the mechanisms to promote health in youth within the section describing the training (in order to avoid possible confusion), and now explain the program theory in more detail, as well as its relevance of its mechanisms and contextual influencing variables, within the part on evaluation design and measures.

“The current study is best conceived as an in-depth refinement of one of the key-influencing factors within this program theory, being the application of motivational coaching and the installation of positive group dynamics by coaches to increase the health outcomes of at-risk youth. Central mechanisms of impact are experiential learning, incremental responsibility-taking, and reflexivity that may, under the right circumstances (e.g., participants feeling safe around others), beneficially influence the health outcomes. These processes and some of the contextual variables may be transferable to the coach training situation regarding health promotion within this same setting of community sports because the local coaches appear to have similar at-risk backgrounds compared to the youth themselves. Therefore, the mechanisms and contextual variables within the program theory provided a good starting point from which interpretations and hidden influential factors could be inferred from the data in the present study. A visual scheme depicting a ‘translation’ of mechanisms of impact, contextual variables, and outcomes is presented in Fig 1.” (page 9, lines 198-210) 

4. Page 6, description of intervention. This paragraph is difficult to read. Perhaps a table could help in seeing which sessions were based on which methods and which strategies? 

We agree that readability of this paragraph could be improved. Initially, we wanted to include the information linking sessions to methods and applications, but we noticed that this leaves the reader with some confusion. Also, it is not essential to include this in detail in light of the aims of the current paper. We now refer to an article describing these aspects in a very detailed manner (also providing tables explaining the links as requested by the reviewer). Within the paper, we changed the paragraph as follows:

“The training program aimed to: (a) increase the awareness and knowledge of coaches on the effects of health behavior on overall health, well-being, and sport performance (e.g., smoking, physical inactivity, poor dietary habits); (b) increase their awareness and knowledge on the mechanisms to promote the health of CYP; and (c) introduce tools and skills to encourage CYP to participate in community sport activities and adopt a healthier lifestyle. To do so, the training program covered topics such as health promotion, healthy living, positive coaching, communication, team dynamics, and conflict. Several strategies were also adopted, including group sessions moderated by one or two tutors with game-based activities, theory and information provision, reflection and discussion exercises, and peer observations. In addition, several individual sessions were planned between and after group sessions. After their first series of four group sessions, each SFD coach had two individual sessions with a job coach, with whom they were already acquainted and had regular encounters regarding their personal (work) trajectories. The aims of these sessions were to encourage elaborated thinking regarding their health status and personal health goals, and the setting of personal action plans regarding their health. At the end of the program, the job coach planned to have at least one follow-up session to discuss the progress of the coaches towards their own healthy living, as well as their concerns or problems in applying skills to promote health among the youth attending the community sports activities. The training aimed to promote change through the application of experience-based and active learning methods, such as raising awareness, guided practice, and skills development, among others. The step-by-step development of the training program, including the links between the methods and their application, is described in detail elsewhere [31].” (pages 7-8, lines 151-172)

5. Methods line 201-205. Include more information on how notes were taken, based on what? Literature? Check lists?

Although there was some general idea of what could be salient (e.g., mechanisms of impact such as group dynamics, or outcomes, such as knowledge or awareness), we tried to observe as open as possible. Therefore, observations were unstructured and unfocused, and it was aimed to write down as much as possible. This also made the observations holistic: everything was taken into account that was perceived to be relevant, allowing an in-depth view on group sessions and interactions between participants. We now provided some more information regarding this approach in text.

“The observer had a general idea of what may be salient but aimed to keep an open mind. The observations were, therefore, unstructured and unfocused, and the narratives were written down with the aim of documenting as much information as possible.” (page 12, lines 239-242)

6. Methods line 207-212. Do the authors know what were major reasons for non-response? Also, I am not sure about the added value of the questionnaires. A scale with only 3 options is not really informative due to the ceiling effect, and they also do not seem to really use the data in the results section (only 2 lines). 

The evaluation questionnaire was asked to be completed immediately after each session. Main reasons for non-response were participants leaving early, and lack of time in completing the forms. Admittedly, the evaluation questionnaire was not that informative and was mainly used as a tool to get feedback from coaches (which was consequently discussed in the next session). We originally chose to include this information within the paper because we wanted to give an uncondensed view on evaluation measures used. But the reviewer is right that it does not add significant content, so we decided to omit it. 

7. Results line 281-283. Can the authors reflect on the low dose that was reached? Why was there a need to lower the intensity? Was there no sense of urgency for the coaches. What were the initial goals of the individual sessions? Why was this deemed not necessary anymore? Similar under ‘fidelity’ line 294. Were the take-home assignments not pretested for wording? 

Indeed, the reviewer is right that in-depth insight is needed here. This is tackled in the section on ‘theory-of-change’, but we do feel we should be more explicit about this and added a paragraph preceding this section. 

“The above findings offer a general overview regarding a diversity of indicators related to feasibility. However, it is important to gain an in-depth understanding of why the training was perceived as acceptable, but was not able to reach all coaches, and the reasons for the apparent disparities regarding dose and fidelity. The following section aims to offer further insights to clarify which training processes led to its success, and identify influential contextual variables by exploring the underlying theory-of-change as perceived by coaches, staff, and other stakeholders.” (page 16, lines 338-343)

Also, some take-home assignments were abandoned because of the observation coaches needed more time and guided instructions to fulfill these. This would have required coaches to spend more time outside group sessions, which was not opted for because of the same reasons why individual sessions seem to have been dropped as explained in the section above. We clarified this as follows:

“The take-home assignments were abandoned because the observation coaches needed more time and guided instruction to fulfill these. The SFD coaches did not opt to spend more time on the assignments outside of the group sessions for the same reasons listed previously regarding the low attendance of the individual sessions” (page 15, lines 318-321)

8. Results line 296-303. The quote does not seem to reflect the statement ‘participants saw the added value to their work’. 

We agree, and omitted this part of the sentence. It is of no added value to the point we wanted to raise regarding acceptability.

9. The title on line 308 is too bold and does not correctly reflect the data, which describes how the participants perceive the mechanisms and the activities that are implemented in the training to produce these mechanisms. The results do not describe how and when the impact of the intervention is reached.

This point relates to one of the major comments (see major comment 3), and we kindly refer to our elaborative answer in response to this comment. We changed the subtitle in our results section (‘towards a supposed theory-of-change: description of perceived mechanisms of impact and influencing contextual variables, see page 16), among other things. 

10. Results line 316-319. Can the authors be more explicit on how ‘the tutor’s openness and willingness to receive feedback’ relates to the ‘coaches/participants motivation’? This seems unrelated. How did the authors see this in the data? In addition, the second quote (line 325) is not really clear to me. I think it misses context to grasp what this person is saying in relation to the analysis of the authors.

The sentence regarding openness being related to motivation links back to the pre-developed program theory that informed this study (a point that we already tackled above). It has no added value here and may even be difficult to understand, and we omitted this link. We removed the quote on the original line 325, as we believe the text is supported enough with the first quote (and we don’t want to include quotes that are ambiguous). 

11. Results line 329. The line seems to suggest a contradiction with the previous statement on openness, but I do not really see this contradiction. Perhaps, because it is not really clear what the main point is from lines 315-319?

We clarified this: 1) by simplifying lines 315-319 and omitting non-necessary information; and 2) by re-formulating line 329.

“Besides openness leading to an increased sense of self-efficacy, it also resulted in some difficulties.” (page 17, lines 363-364)

12. Results line 345-347: anecdote is not clear, please specify what is meant with ‘pushing to the limit’ and ‘unconditional’. 

We changed this sentence, as follows: 

“One staff member called it ‘a door swinging both ways’. This member made the parallel with a project targeting at-risk youth and, as such, tried to demonstrate the importance of the unconditional nature of activities, also when targeting coaches.” (page 18, lines 377-379)

13. Quote 376-380. The quote does not seem to address involvement or engagement but rather satisfaction with the program. 

The problem there is that there were multiple ideas involved in one quote. We kept the sentences related to the point we wanted to make, namely the fact that breaks in between activities were perceived as necessary to carry on and keep engaged with the training. We re-structured this section (see page 19). 

14. Page 13 contains three quotes, I think one quote would be sufficient and makes it easier for the readers to see the main point of the authors. 

We agree with the reviewer that all three quotes may not be needed. We kept two quotes. One illustrates the idea explained by staff that coaches become aware of the importance of setting an example, being an attitude shift observed throughout training. The second quote is important in a sense that staff perceives this role modelling to be a powerful lever, possibly related to bringing about change among at-risk youth in a later stage (see page 20). 

15. Quotes 421-427. Two quotes: to what extent do they address involvement and engagement. I do not really see the link. Moreover, it would be helpful to explicate what the mechanism is that is at work here.

We agree that we could illustrate more what is the ‘realist thread’ that is underneath these ideas. We now explicated this more in the text, as follows:

“At the same time, however, a discrepancy arose between having to act as a role model and feeling able to be one. For example, when the behavior of current coaches was in contradiction with the behavior to be promoted among youth (e.g., smoking vs. preventing youth from smoking). This discrepancy highlights the possibility that personal vulnerabilities can lead to coaches having uncertainties or doubts of whether they will be able to live up to being a role model; thus possibly hampering the link between learning through training and a shift in thinking or the awareness of being a role model for at-risk youth.” (page 20, lines 442-448)

16. Quotes 435-445. Again, one quote will be enough. In addition, the quotes are a bit vague and I do not really read in these quotes the main point that the authors are making in line 429-433.

We agree that three quotes are not needed, but kept two: one referring to the experience of a coach (this one was imputed from elsewhere in the text as we felt it better suited the idea we wanted to transfer) and one relating to the experience of one of the staff members. Also, we re-wrote the sentence on the original lines 429-433 in order for it to better reflect the idea of coaches being engaged to monitor their own coaching behaviour and, as a result, experience an increased sense of self-efficacy regarding their coaching skills. 

“Also, engagement was perceived to result in increased self-monitoring of current coaching practices. As a result, some SFD coaches specifically mentioned that they had become aware of what they had already done and achieved, leading to increases in overall self-efficacy.” (pages 21-22, lines 473-476) 

17. Results 462-466. Results section ‘Barriers that moderate the intervention impact’ is not addressing mechanisms. As this part of the results is directed at identifying mechanisms, is this a relevant point to add? In addition, the authors do not seem to do a Context-Mechanism-Outcome analysis, so they would be unable to identify barriers that moderate the impact. 

As is hopefully made clear already, our intent with the section (and each and every subsection) is to offer insight into the training’s theory of change by exploring the training processes related to possible success, and identifying influencing contextual variables. Within every subsection, the experiences related to such training processes are discussed, also in light with the outcomes to be named and influencing contextual variables. We re-phrased the paragraph on the original lines 462-466 so to increase clarity.

“However, staff also mentioned variables external to the training that seemed to hamper its implementation or impact. These were related to the characteristics of coaches as being a vulnerable group themselves, such as suffering from low levels of self-esteem and having (a history of) (mental) health problems. In addition, language problems were apparent in one coach and appeared to hamper their training implementation.” (pages 22-23, lines 497-501) 

18. Results section ‘group dynamics and relations’. The authors conclude that this mechanism is not vital, but they do mention it as an element in the abstract. I would leave it out if it is not important. Also, it is confusing that the results section talks about mechanisms, but then also mechanisms that were not playing a role. 

This could be confusing, but we notice a misunderstanding, possibly due to our wordings of the respective paragraphs. Group dynamics is an essential process related to beneficial training outcomes. A related process is the building of bonds of trust between tutors and staff as well as coaches. Those two processes are described under the same header. With regard to the establishment of positive group dynamics, we noticed it, although being perceived of as an essential process, to be easily attained under the circumstances that there were already well-established relations between coaches and staff beforehand. We re-phrased some wordings to make this clearer.

“Coaches and staff recognized the benefits of promoting group dynamics for the training to be successful; a process that was easily established within the studied context, in which everyone knew each other adequately well beforehand.” (page 23, lines (512-515)

19. Results 493-496, is this not more a mechanism of involvement and engagement? 

There was some rehearsal and not all sentences were needed to make our point. We re-phrased some sentences.

“In addition, the staff highly appreciated the inviting attitude of researchers and tutors regarding the co-creation and co-delivery of the intervention. This process resulted in the growth of constructive bonds and relationships between the researchers and the staff of the particular SFD organization, and seemed to contribute to a positive learning environment. The resulting learning effects, as mentioned by staff, was that such co-working opportunities resulted in deepened reflection regarding their current practices and ways to do better. ” (pages 23-24, lines 522-527)

Also, we moved one quote to the section on acceptability (feasibility), because we noticed, upon revision, that it was stated too general to be included in the subsection on group dynamics and relations. Nevertheless, it gives voice to the perception of staff appreciating the training as a whole, particularly because it was supportive and not perceived as extra work. 

20. Results ‘setting personal health goals’. Can the authors be more explicit that they talk about the health goals of the coaches themselves and not the participants of the SFD program (at-risk youth). I found this confusing throughout the paper. I addition, it is also concluded that this program element did not seem to work. It would be nice if they could offer a more critical analysis of this in the discussion.

We agree that this paragraph should be re-written in a more clear and concise way. In particular, we noticed that we did not make clear which essential process (or mechanism) did work, being the awareness on and increased monitoring of own health behaviour by coaches. We re-named this process as such, as this was the main idea that we wanted to share (see page 24). 

And true, we did not notice coaches to set short-term goals in changing their behaviour, nor did we observe an actual impact on the health behaviour of coaches. This finding is discussed in detail within the discussion section (see pages 19-20, lines X-X)

“It is important to note here that, based on our findings, the training did not easily facilitate changes in the behavior of coaches, which is an important prerequisite for social prompts to occur. Coaches did not report changes in setting personal health goals themselves nor increased efforts towards accomplishing health goals or adopting healthy behavior. Those who reported change (e.g., a few coaches mentioned changes in their dietary intake or smoking behavior) mentioned the diminishing of the training effects on their behavior after a while. These findings were consistent with the staff observations that stressed that the training did not lead to observable, long lasting changes in health behavior despite challenging the coaches to question their health status and risky health behavior. However, these findings were not entirely remarkable and were consistent with theoretical ideas on behavior change. Process theories on behavior change, for example, delineate behavior change as a time-consuming endeavor that proceeds through different ‘stages’ [43,44]. These stages differ between theories, but there are also communalities among theories. For example, it is commonly theorized that individuals first proceed through a stage of awareness and knowledge building before they express an intention to change their behavior, and for the better. In parallel, the training may ultimately be effective in changing the awareness and knowledge of participants, but for it to have a lasting impact on the behavior of coaches, more actions may be needed. Here, we advocate more intense individualized guidance and feedback of coaches, which could not be obtained from a group-based format. In addition, the time frame may need to be lengthened, so that long-term follow-ups are possible, and the opportunities to transfer skills to real-life settings are increased.” (pages 28-29, lines 636-656) 

21. Results 527-536: this paragraph is vaguely formulated, could the authors provide examples of what they mean for example ‘necessary vocabulary’ and ‘implement new actions in practice’. 

We revised this section and added some illustrations to increase clarity (see page 25).

22. Quote 542-545: is this quote about the SFD program for at-risk youth? Does the coach say that he keeps track of the emotional tank of the youth? Because that would not fit the ‘mechanisms underlying the training for coaches’. (see also major issue 2).

This is rightfully noted, and this quote should not be placed in this section. We transferred the quote to the section on ‘building involvement and engagement’, and introduced it as follows:

“Several other outcomes that were enumerated and linked to building involvement and engagement included an increased awareness, an improved knowledgebase, and a stronger motivation regarding (new) health promotion practices. One example is the learning of new concepts that gave coaches the language to discuss and explain the things that they were already doing (e.g., coaches already knew the importance of not putting too many demands on youths, but the concept of ‘an emotional tank’ clarified the concept further, and gave them the language needed to discuss these issues with youths as well as with staff and fellow coaches), and contributed to the motivation of coaches to continue their coaching, for the sake of the health and well-being of youths.

“I actually always keep track of the emotional tank. I give a lot of compliments because it helps. When something is fun, then they can handle it better. It helps. If you only say something when they are doing something bad, but never when they are doing something right, then the children’s reservoirs are emptied.” (Coach 2) (page 21, lines 458-471)

23. Quote 547-550: I am not sure in what context to place this quote. Who is ‘they’? Is this about the coaches or the at-risk youth?

Yes, we added this information to the quote.

“They [the coaches] were also thinking about who they were going to put together in a group and who not. And now thinking about it, there was also a coach who said ‘hmm, that game in that neighborhood, we’re not going to do that because of what we saw last week, that if we play that game in that neighborhood then we are actually making everyone feel uneasy.” (Staff member 2)(page 26, lines (579-583)

24. Results 552-554: is this a complete paragraph? What message are the authors trying to get across? 

We agree it should be integrated with information above, further specifying the process of reflection and discussion. We removed the information to the specified section (see page 25). 

25. Results 556-560: is this not better matching ‘involvement and engagement’.

We removed this information to lines X-X of the same section, better clarifying the contextual variables that are explained here.

“The data revealed the fostering of discussion and reflection on current functioning to be one of the most important processes related to the training outcomes, as mentioned by the coaches and staff during the observations and interviews. Staff and coaches also mentioned the importance of reflection being well guided (e.g., through imagery, case-examples, via step-wise instructions, etc.). Unguided reflection (e.g., open questions) was often perceived as being intrusive and threatening by both coaches and staff.

An important contextual variable that facilitated the success of reflection was the background of coaches themselves possessing relevant practice-based knowledge. This made reflection specific, which contributed to this process being a valuable learning tool. Another important condition was the timing of reflection, which was related to the creation of a safe environment and bonds of trust. In addition, reflection in the absence of safety and trust appeared to raise the threshold regarding engagement and led coaches to being less involved and less responsive during sessions.” (page 25, lines 551-563)

26. Discussion 637: context elements that impact the intervention. Is this a new topic? Is this also part of the results section, I haven’t seen these. Or are they the same as mechanisms? This is a very short paragraph that needs elaboration. And I wonder if this is actually a result from this current study? Again, no COM configurations were analyzed. 

We adapted the discussion section in a significant way, also in line with a more clear and consistent presentation of our results. As is hopefully made clear now, one of our aims was to explore the feasibility of the training for coaching through a realist lens. This provides us with in-depth insight on the inner workings of the training, as well as the contextual variables that ought to be set in place in order for the processes to result in preferred outcomes. We repeated this approach in the discussion section, followed by an in-depth discussion of each of the important processes and outcomes, also in light of the contextual variables facilitating or hampering training and/or outcomes. 

27. Discussion paragraph 646-669 could be deleted, does not seem to add anything new.

We agree with the reviewer that this paragraph stood alone, and was not elaborative enough to be of value. We re-wrote the point we wanted to make, namely the value of the general appreciation and outcomes of the training, and the change we wanted to observe but was not brought about, and linked it to relevant theoretical insights and model. As such, we provided interesting avenues for future work with alike programs in the context of sport-for-development. 

“Overall, our findings showed that the training was suitable and well accepted by coaches who expressed positive responses to the training, i.e., it increased their awareness of health and their actions as role models for at-risk youth. This overall finding has great importance. One of the key change mechanisms of improved health among at-risk youth using sports as a lever is their observation and learning through what is respected by others and in particular with regards to what the coaches are doing [34,35]. This process of ‘vicarious learning’ is most profoundly described in the Social Cognitive Theory (SCT) of Bandura [39]. 

Based on SCT, several pathways of learning could be presumed. The first is the most direct pathway: coaches may give information and instructions on how to change behavior. For example, a coach may explain the negative effects of smoking on overall health. The second pathway involves the health behavior of the coaches, which may motivate youth to behave in a similar way because of the benefits they perceive or hear the coaches express. For example, a coach expressing feeling fitter after having quit smoking may motivate youth to also quit smoking. Thirdly, coaches may serve as a social prompt for CYP to perform healthy behavior amongst different alternatives. For example, a coach may set an example by drinking water and eating fruit during sport activities. Despite the pathways differing, they are all deemed important. While the first two pathways may require more intentional, rational thinking (i.e., by gaining knowledge and changing attitudes), the latter pathway is likely to occur in a more automatic, unintentional manner. Within the Elaboration Likelihood Model of attitude change of Petty and Cacioppo [40], this difference is referred to as the central and peripheral route. The central route consists of thoughtful consideration of the arguments (ideas or content) of the message. The peripheral route occurs when the listener decides whether to agree with the message based on other cues, such as the bond or perceived power of the person delivering the message. Research suggests, albeit mostly in college students, that a few variables may influence the extent to which people are more likely to be convinced by contextual cues rather than by the message itself. Examples include not being motivated by the content of the message (e.g., having no interest or seeing no benefit), not being able to think about the message (e.g., because of being distracted by other things), or having to think about a message that is difficult to comprehend [41,42]. Although it has not yet been investigated, at-risk youth are likely to be more sensitive to health messages delivered by coaches because of their status, trustworthiness, and perceived power, than through the content of the message itself. Therefore, the social prompting of health messages may seem to be an important route of transfer between coaches and at-risk youth, and is thus an assumption worthy of further investigation.” (pages 27-28, lines 602-635) 

28. Discussion line 670: authors talk about ‘promising outcomes’ but these were not part of the results section. Further on they talk about data that pointed out that certain factors hamper intervention delivery, but these were marginally discussed and not very thoroughly elaborated upon in the results. 

We agree with these comments, and upon revision, these points are made more clear. The outcomes are discussed on pages 27 and further (see previous comment), and contextual variables are discussed in light of the processes that we found to be relevant in terms of the outcomes it was perceived to bring about (see also comment 27). The changes we made are that extensive, that we opted to do not indicate these below. All changes to the discussion section are indicated in-text.

29. Limitations line 711-713: please explain how this influences the findings and their validity. Do the authors feel that valid conclusions can be drawn from the interviews? And to what extent should we be careful drawing conclusions?

We noticed that we created unnecessary confusion about the validity and reliability of our findings. Also, the suggestion about using video-coded interactions does not add much to our study, so we omitted this. 

“Some of the interviews with SFD coaches were difficult, wherein some struggled to elaborate on questions regarding supposed mechanisms, context factors, and outcomes. We used triangulation to solve this issue, combining multiple observers and methods. We are, therefore, confident that our results are valid and reliable.” (page 34, lines 782-786)

30. Limitations line 732-735: paragraph needs elaboration.

We changed this paragraph, as we omitted quantitative-descriptive data because these were not of added value. The idea of the study not being an effect evaluation is evident, and not a necessary remark to be given, and, therefore, we also omitted it. We changed the paragraph as follows:

“Fifth, we conducted a process evaluation and were interested in the feasibility and theory-of-change underlying the feasibility, i.e., processes at work that impacted training outcomes, under specific circumstances. We used a qualitative research design to gain in-depth knowledge regarding the feasibility and, in particular, the experiences surrounding the training, underlying processes, and the complex links with outcomes and associated contextual variables. Quantitative or mixed-method designs can be used in future studies to test these pre-hypothesized pathways of impact, and contextual moderators. In future studies, data should be collected at multiple time points to capture the effects of the training over time.” (page 35, lines 803-810)

 

Reviewer #2: This study about an intervention for SFD coaches is very interesting because it is a process evaluation, and the execution of multiple methods. The intervention has the goal to inform and train SFD coaches to coach CYP regarding their health related behaviour. The intervention focuses on knowledge acquisition for unhealthy behavior and mechanisms to promote health, and coaching skills. Coaches appreciated evaluation and reflection most and the fact that they were confirmed in their qualities, and became aware of their role and position.

This is an excellent summary, and the reviewer indeed raises some of the central processes of impact observed throughout the data. We would specifically like to thank the reviewer for the kind appreciation of the study. 

Method: The chosen method is very suitable and extensive to evaluate the intervention. It brings above the experiences which are necessary to implement or adjust the intervention. However, with the description of the method tells us that this study will reveal how the intervention was experienced by the coaches. The title is misleading due to the focus on the improvement of health of youth and would be better with a focus on coaches, who have to affect the health related behavior of youth.

We suppose the reviewer here refers to the description of the method in the abstract, which we indeed changed into the following:

“In this study, we evaluated the feasibility of such a training program for SFD coaches using process evaluation from a realist perspective, and views from multiple stakeholders, among other sources. We also clarified the inner workings of the training and investigated how context shaped the training outcomes.” (page 2, lines 25-29)

And indeed, we thank the reviewer for noticing the different kind of target that is taken in the original title. We changed the title into “Evaluation of a program targeting sports coaches as deliverers of health-promoting messages to at-risk youth: assessing feasibility using a realist-informed approach”; as referring to coaches indeed better reflects the scope of the current paper. The short title then became: “Feasibility of training programs targeting sports coaches as deliverers of health promotion messages”.

Data: The data supports the conclusion partly because the conclusion is too general and not critical enough.

The 5 points were clear and helpful for other interventions. Point 4 triggered because I think it was about awareness in stead of setting personal goals.

The intervention was good but not fully developed. This study gives the opportunity to develop the intervention further. However, a critical analyse is therefore necessary. As a reader, I think the coaches appreciate the reflection, discussion and theory about awareness and knowledge about promoting health. They already have the knowledge about unhealthy behavior, and due to the focus of the first sessions on this theme, presence can be lowered. It would be helpful if new ideas would be presented to create more presence. And if individual sessions are desired amd what should be the focus of these sessions.

We thank the reviewer for these excellent points, and below we refer to each of the elements separately.

- Conclusion: Indeed, this point has also been raised by reviewer one, and we now changed the conclusion as to better reflect the main data. We follow the suggestion of the reviewer, and wrote it in a realistic and critical manner.

“A realist-informed process evaluation was deemed the most suitable approach to determine the feasibility of training to increase the awareness and ability of coaches to transfer health prevention messages to at-risk youth. We used the MRC guidelines as a framework for our analysis. We were also interested in clarifying the underlying mechanisms of the impact and context factors that led to the supposed training effects. A case study was undertaken allowing the collection of views from multiple stakeholders and from a variety of sources. A safe learning environment seemed to contribute to a ‘right mind-set,’ which facilitated the learning process among peers and tutors with whom the coaches felt connected. An increased health awareness and sense of responsibility to act as a role model for at-risk youth were among the main outcomes, and was reached through an increase in self-confidence, and an improved sense of critical self-reflection and self-development. These outcomes were triggered through the above-described processes under certain conditions. In addition, several variables, such as a precarious life history or living conditions, mental health issues, or low educational skills, may hamper the processes and outcomes. This study offers valuable insights into the processes and appropriate circumstances of SFD training that may prepare coaches to effectively deliver health prevention messages. This may inform intervention developers and policy makers to make more sensitive and suitable choices in the setting up and implementation of health prevention programs through sports using coaches as deliverers of those messages. Additionally, programs such as the one used in the present study are very unlikely to directly impact health behavior, but rather set the stage for further individualized actions to be undertaken. Therefore, programs should be assessed and monitored using the intermediate outcomes most at stake. This includes motivation, and even more proximate indicators, such as awareness, self-efficacy, sense of self-reflection, and a sense of responsibility as good candidates.” (pages 35-36, lines 813-836)

- Point 4: We agree, and therefore, within the results section, we named this process ‘personal health monitoring’, indicating an awareness of the importance to register own health behaviour. Indeed, this is not about the actual goal setting, or even achievement of goals and the original title was misleading in this respect. Within the results section, we elaborate on this based on the data (see subsection ‘personal health monitoring’ on page 24), and within the discussion section, we discuss this in-depth and refer to theoretical concepts and models that help us to understand these experiences (see pages 29-30 and further on). We do not provide all changes below, but all amendments are visible in-text.

- Critical appraisal of the intervention: Indeed, the current study aims were to discuss feasibility and provide researchers and policy makers with ideas on how to implement alike initiatives within similar contexts. The remark on the need to complement the training with individual sessions is correct. Also correct is the notion that this aspect was not the main focus of the training. Individualized sessions were aimed for, but due to contextual circumstances, these were not implemented, or only in a limited, non-structured manner. As a consequence, we can only give recommendations to implement these, especially when aiming for long-term behavioural change. We discuss this in a new paragraph within the discussion section (see page 29, lines 676-676). Also, by performing a realist-informed process evaluation, our study provides plenty of ideas and insights on how to make a coach training within this context more successful. We do acknowledge that the added value of our approach may not have been visible as such. We revised most paragraphs explaining the inner workings of the training impacting outcomes. We do also provide insight into the contextual conditions that have to be at place in order for processes to have their impact. We are convinced that our revision helps to the critical analysis of the training. One example is the issue of presence tackled by the reviewer. This is discussed in light of the first process, being the establishment of a safe and trustworthy environment. In describing this process in depth, we offer several, often practical, suggestions on how to target coaches in order for them ‘feeling safe enough’ to participate in the training. At the same time, we also discuss important contextual variables, such as a too professional out-look of the training hampering safety experiences among coach. These contextual boundaries should be considered when developing and implementing alike initiatives. The paragraph reads as follows:

“The first finding relates to the type of environment that was created at the start of the training and continued throughout the program, i.e., the establishment of a safe and trustworthy environment for participation in the intervention and for the personal development of SFD coaches. Our study showed that a sense of safety reduced the doubts and insecurities of coaches, and produced an environment that was conducive to learning and which did not punish individuals for not participating within the training or when missing one or more training components. This idea of ‘psychological safety’ is consistent with other scholarly publications on the conditions that need to be met for people to be motivated to perform, i.e., to learn and participate in the training in this respect. Based on Self-Determination Theory (SDT) [45], these studies highlighted that the creation of a social context that motivates participants or athletes to the degree in which they feel autonomous and sufficiently free, positively affects their level of intrinsic motivation and enjoyment (e.g., [46]). Indeed, the friendly attitude and openness that both the SFD coaches and staff expressed towards intervention deliverers was related to their increased awareness of the need to discuss the topics and intrinsic motivation to proceed with the intervention. However, this appears to be a precarious situation. It requires amicable, supportive, and inclusive but also professional relations between SFD coaches and staff as well as with intervention deliverers. Excessive professionalism (e.g., requiring the completion of assignments, strictness regarding punctuality, etc.) may create the perception of an unsafe environment and hamper the confidence of coaches to learn.” (page 30, lines 677-696)

Analysis: Analysis were suitable and good.

Thank you.

Availability of data: This study obtained a lot of useful information. The researchers declare that all data is available. However it would be useful if we get an insight in the result of the thematic analyse, questionnaire results. It is not clear if these results will be available in an appendix.

Also in line with the recommendations offered by reviewer one, we decided to omit the ‘questionnaire’ and respective ‘results’ (see reviewer 1; minor comments, comment 6). The questionnaire was not informative, and mainly used as a tool to get feedback from coaches (which we consequently discussed in every following session). We originally chose to include this information within the paper because we wanted to give an uncondensed view on evaluation measures used. But given the fact it does not add significant content, we decided to omit it now. Data from the thematic analysis are provided within the text itself, as the quotes itself reflect unchanged data (experiences) from the observed. 

English: There were a few minor elements such as a missing space in the abstract.

Thank you, we now noticed that. We meticulously revised the paper and eliminated all spelling mistakes. Please also note that we used a professional editing service and, as such, the paper should be written in an intelligible way, and in clear language. 

A few questions, which can be described in the article:

- The introduction focuses on access to services, but isn't it about health literacy?

We indeed noticed that the introduction may be misleading in some respect. We wanted to describe how at-risk youth run a higher risk of health issues. Therefore, in one of the paragraphs, we focus on inequity in access to health services as one of the important causes in these health differences. We wanted to make this point and propose SFD programs to offer one of the possible solutions in reducing health inequity, among other initiatives to be taken. One of these may rightfully be the improvement of health literacy, as noted by the reviewer. But since this is not the focus of the paper, we decided to make our point more clear, and write more hypothetically, leaving enough space for other options in reducing inequity to be important. We made the following changes, in the abstract and introduction section.

“Initiatives that improve the access to and adoption of health promotion messages are important undertakings, e.g., sport. Sport-for-development (SFD) programs are seen as valuable delivery tools, in which coaches are used as change agents to increase health awareness and behavior among at-risk youth.” (page 2, lines 20-23)

“Inequities in long-term health can be reduced not only by improved access to health promotion and prevention services but also by the tailoring of these services to the circumstances of CYP, among other initiatives [8].

Sport-for-development (SFD) may provide a setting through which health promotion and prevention messages can more easily reach and be adopted by hard-to-reach populations.” (pages 3-4, lines 62-66)

- Only male participants and from Bruges. Do you think the intervention is suitable for women and somewhere else?

We have no indication that gender would be an important variable pertaining to the successes of such a training. But the question is helpful and right, because we do want to raise the question on suitability of programs in the sideline of the current paper. This study highlights the importance of triggering essential processes that may impact training outcomes. Of course, whether these processes may deliver intended training outcomes is dependent upon contextual factors. Some may hamper outcomes, other may rather facilitate training outcomes. Therefore, suitability should never be assumed beforehand, and this study gives insight into measures to be taken and questions to be asked when wanting to implement and evaluate alike training elsewhere. We added this note to the paper:

“The main processes observed appeared to be essential for the successful training of the coaches, however, precaution is warranted regarding the impacts of such training. These impacts may be limited by the presence of contextual variables, such as the level of (health) literacy, mental health, and the living conditions of coaches. As observed in the present study, context may also facilitate training among coaches and staff with already established bonds of trust. This study offers insights into measures to be taken when implementing and evaluating similar training programs elsewhere.” (page 33, lines 763-769)

- How did you decide that fidelity can be marked as good? The same question arised when you state that the intervention is suitable and accepted.

Indeed, we were possibly overly positive when noticing that session content was mostly delivered as planned. But we do agree with the reviewer that we cannot state this in general, and we changed the wordings. 

“Overall, the session content was delivered as planned, but we also observed disparities. Activities and exercises were omitted to a minor extent. In most cases, however, task assignments that were scheduled between sessions were not fulfilled.” (page 15, lines 308-310)

Also, the term ‘suitable’ when referring to ‘acceptability’ may not be preferred, as this more likely refers to the training as a whole, and we do want to make a critical appraisal. We re-phrased the information under the subsection ‘acceptability’ as follows. We also added one quote to support our statements.

“Participants perceived the intervention as important, satisfactory, and educational. The intervention was not perceived as something additional, but rather as something that supported their current work.” (page 15, lines 324-326)

- Is the timeframe large enough, or should it be lengthened so the staff can follow coaches their development and train them in real life settings?

Yes, indeed, correct remark, and we added this idea to our discussion section. 

“In addition, the time frame may need to be lengthened, so that long-term follow-ups are possible, and the opportunities to transfer skills to real-life settings are increased” (page 29, lines 654-656)

- Which staf member of coach stated the quotations? It would be helpful if everyone has a number, so it is visisble that the quotations do not stem from 1 or 2 persons.

We agree that this information was indeed missing, and we added it. 

  

6. PLOS authors have the option to publish the peer review history of their article (what does this mean?). If published, this will include your full peer review and any attached files.

Do you want your identity to be public for this peer review? For information about this choice, including consent withdrawal, please see our Privacy Policy.

Reviewer #1: No

Reviewer #2: No

 

Journal Requirements:

Please ensure that your manuscript meets PLOS ONE's style requirements, including those for file naming. The PLOSONE style templates can be found at

We used a professional editing service that helped us with preparing the resubmission in order for it to meet PLOS ONE’s style requirements.

1. Please include additional information regarding the survey or questionnaire used in the study and ensure that you have provided sufficient details that others could replicate the analyses. For instance, if you developed a questionnaire as part of this study and it is not under a copyright more restrictive than CC-BY, please include a copy, in both the original language and English, as Supporting Information. Moreover, please include more details on how the questionnaire was pre-tested, and whether it was validated.

We omitted the ‘questionnaire’ and descriptive results are not reported, as suggested by one of the reviewers.

2. When reporting the results of qualitative research, we suggest consulting the COREQ guidelines: http://intqhc.oxfordjournals.org/content/19/6/349. In this case, please consider including more information on the number of interviewers, their training and characteristics

We indeed consulted the COREQ guidelines while preparing the initial submission, and again consulted it to check whether everything indicated in the list was present in the paper. Indeed, we agree that information regarding the interviewers was missing. We included this, as follows:

“All interviews were held in person by the third author in a private room at the SFD setting. This researcher had content-related experience which gave breath to the data collection. Due to this experience, a self-reflective stance was adopted during interviewing, and, having extensive methodological expertise, a general openness and curiosity about the interviewees’ experiences was established.” (pages 12-13, lines 248-253)

“Two researchers (EL and KVDV), both of whom had different content-related and methodological expertise, collected the data.” (page 13, lines 266-267)

3. Please ensure that all statements made (for example, the ones at lines 86-87 of the Introduction) are supported by relevant citations .

We carefully checked this. Indeed, the respective sentences missed references. We adapted it as follows:

“In addition, the effectiveness of the interventions depend on the extent that coaches manage to deliver appropriate knowledge, install appropriate attitudes, and apply appropriate skills regarding the health promotion activities to be delivered [25]; which is especially challenging within SFD. SFD coaches also introduce a range of different skills and personality profiles into SFD initiatives, given the different roles and skills required to coach at-risk youth. Some of these roles include being a trustful friend, a liaison officer between the youth and youth organizations, and a technical coach to develop the sportive capabilities of some individuals [26,27]. In Flanders, Belgium, organizations often combine these different roles by recruiting different profiles. For example, such scattered workforces can consist of mid- to highly educated coaches that possess sportive or pedagogical degrees (or both) as well as experienced experts that often do not have degrees and have low educational skills.” (pages 4-5, lines 81-92)

4. Please include your tables as part of your main manuscript and remove the individual files. Please note that supplementary tables (should remain/ be uploaded) as separate "supporting information" files

We adapted this. 

 

References

Gilmore B, McAuliffe E, Power J, Vallières F. Data analysis and synthesis within a realist evaluation: toward more transparent methodological approaches. Int J Qual Methods. 2019; 18:1–11.

Van der Veken K, Lauwerier E, Willems S. How community sport programs may improve the health of vulnerable population groups: a program theory. Int J Equity Health. 2020a.

Van der Veken K, Lauwerier E, Willems S. “To mean something to someone’”: sport-for-development as a lever for social inclusion. Int J Equity Health. 2020b; 19:11.

---

## [Decision Letter · Decision Letter 1]

15 Jul 2020

Evaluation of a program targeting sports coaches as deliverers of health-promoting messages to at-risk youth: assessing feasibility using a realist-informed approach

PONE-D-20-01877R1

Dear Dr. Lauwerier,

We’re pleased to inform you that your manuscript has been judged scientifically suitable for publication and will be formally accepted for publication once it meets all outstanding technical requirements.

Kind regards,

Elena Cavarretta, M.D., Ph.D.

Academic Editor

PLOS ONE

Additional Editor Comments (optional):

Reviewers' comments:

Reviewer's Responses to Questions

**Comments to the Author**

1. If the authors have adequately addressed your comments raised in a previous round of review and you feel that this manuscript is now acceptable for publication, you may indicate that here to bypass the “Comments to the Author” section, enter your conflict of interest statement in the “Confidential to Editor” section, and submit your "Accept" recommendation.

Reviewer #1: All comments have been addressed

Reviewer #2: All comments have been addressed

Reviewer #3: All comments have been addressed

2. Is the manuscript technically sound, and do the data support the conclusions?

Reviewer #1: Yes

Reviewer #2: (No Response)

Reviewer #3: Yes

3. Has the statistical analysis been performed appropriately and rigorously? 

Reviewer #1: Yes

Reviewer #2: (No Response)

Reviewer #3: Yes

4. Have the authors made all data underlying the findings in their manuscript fully available?

Reviewer #1: Yes

Reviewer #2: (No Response)

Reviewer #3: Yes

5. Is the manuscript presented in an intelligible fashion and written in standard English?

Reviewer #1: Yes

Reviewer #2: (No Response)

Reviewer #3: Yes

6. Review Comments to the Author

Reviewer #1: (No Response)

Reviewer #2: Thank you for your revisions. It is a new article and much more realistic and helpful for practice.

Compliment for the changes.

Reviewer #3: (No Response)

7. PLOS authors have the option to publish the peer review history of their article (what does this mean?). If published, this will include your full peer review and any attached files.

Reviewer #1: No

Reviewer #2: No

Reviewer #3: No

---

## [Editor Report · Acceptance letter]

21 Aug 2020

PONE-D-20-01877R1 

Evaluation of a program targeting sports coaches as deliverers of health-promoting messages to at-risk youth: assessing feasibility using a realist-informed approach 

Dear Dr. Lauwerier:

I'm pleased to inform you that your manuscript has been deemed suitable for publication in PLOS ONE. Congratulations! Your manuscript is now with our production department. 

Kind regards, 

on behalf of

Dr. Elena Cavarretta 

Academic Editor

PLOS ONE